# COMI: Coarse-to-fine Context Compression via Marginal Information Gain

**Jiwei Tang**[1,2] **Shilei Liu**[2] **Zhicheng Zhang**[1] **Yujin Yuan**[2] **Libin Zheng**[3*] **Wenbo Su**[2]
**Bo Zheng**[2*]
[1]Tsinghua University   [2]Future Living Lab of Alibaba   [3]Sun Yat-sen University
tangjw24@mails.tsinghua.edu.cn   zhenglb6@mail.sysu.edu.cn
bozheng@alibaba-inc.com

## Abstract

Large Language Models (LLMs) have demonstrated exceptional capabilities across diverse tasks. However, their deployment in long context scenarios remains hindered by computational inefficiency and information redundancy. Context compression methods address these challenges by significantly reducing input length and eliminating redundancy. We propose **COMI**, a coarse-to-fine adaptive context compression framework that jointly optimizes for semantic relevance and diversity under high compression rates. We introduce *Marginal Information Gain (MIG)*, a metric defined as the relevance of a unit to the input query minus its semantic redundancy with other units, guiding the compression process to prioritize information that is both relevant and low redundant. The framework operates in two stages: (1) **Coarse-Grained Group Reallocation**, where the context is partitioned into groups and dynamically assigned compression rates based on inter-group MIG, ensuring compression budgets align with information value distribution; and (2) **Fine-Grained Token Merging**, where tokens within each group are fused via an intra-group MIG-based weighting mechanism, thereby preserving key semantics while avoiding the accumulation of redundancy. Extensive experiments across question-answering (*e.g.*, NaturalQuestions, 2WikiMQA, HotpotQA and NarrativeQA), summarization (*e.g.*, MultiNews) with various backbones (*e.g.*, LLaMA-2-7B, Qwen2-7B) show that COMI outperforms existing baselines by a large margin, *e.g.*, approximately 25-point Exact Match (EM) improvement under 32x compression constraint with Qwen2-7B on NaturalQuestions[1].

## 1 Introduction

Large Language Models (LLMs) achieve exceptional performance across a wide range of Natural Language Processing (NLP) tasks (Yang et al., 2024; DeepSeek-AI et al., 2025; Team et al., 2025; Lv et al., 2025; Zhao et al., 2025b; Liu et al., 2025a). However, recent advances in prompting techniques, such as Retrieval-Augmented Generation (RAG) (Lewis et al., 2020), inevitably increase input length, introducing two key challenges when deploying LLMs in long context scenarios: (1) computational cost, as the quadratic complexity of the attention mechanism in Transformer (Vaswani et al., 2017) leads to inefficiency with long sequences; and (2) information redundancy, where the presence of redundant content degrades model performance (Jiang et al., 2024; Ge et al., 2024; Liu et al., 2024b; Tang et al., 2025a;b).

Context compression emerges as a promising solution in the NLP community to address these challenges by significantly reducing input length and eliminating redundancy. Existing prompt compression methods mainly fall into two categories: task-agnostic context compression methods (Xu et al., 2024; Pan et al., 2024; Ge et al., 2024; Tan et al., 2024; Cheng et al., 2024; Zhang et al., 2025; Ye et al., 2025; Li et al., 2025; Liao et al., 2025; Tang et al., 2025b) and task-aware context compression methods (Cao et al., 2024; Jiang et al., 2024; Zhao et al., 2025c; Tang et al., 2025a; Fang et al., 2025; Liskavets et al., 2025). LLMs typically allocate attention to only a small subset of

---

*Corresponding authors
[1]The code will be available at https://github.com/Twilightaaa/COMI

query-relevant tokens (see Figure 1a). Consequently, task-agnostic methods that compress context without considering the input query inevitably lose or dilute relevant information, especially at high compression rates.

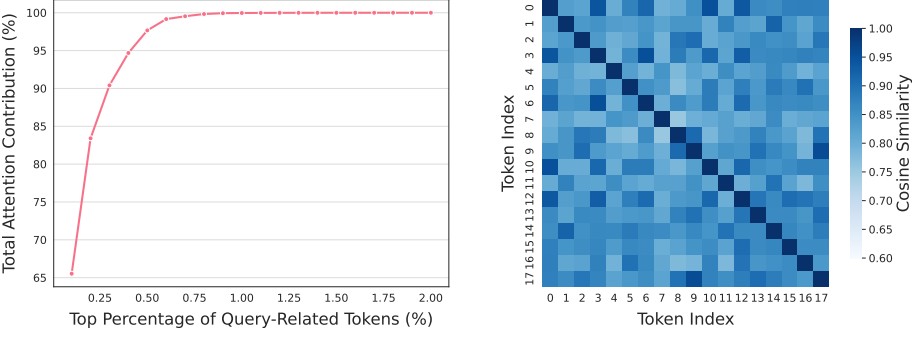

(a) Attention Weight Contribution.    (b) Top Query-Related Tokens Similarity.

Figure 1: Analysis of Attention Distribution and Similarity of Top Query-Related Tokens. (a) Only a small number of tokens related to the query occupy a large proportion of the attention weight allocation; for example, the 0.75% most relevant tokens occupy 99% of the attention weights. (b) These query-related tokens are highly similar to each other, with the lowest similarity exceeding 0.6. In contrast, task-aware methods generally include query in the compression process, preserving information according to relevance via query-guided strategies such as merging, deletion or summarization. However, existing task-aware methods rely solely on relevance as a criterion for compression, ignoring the inherent redundancy in natural language (Shannon, 1948). This leads to the retention of *highly similar* relevant content, with high redundancy (see Figure 1b). Such over-similarity can mislead the LLM into producing erroneous outputs (relevance does not guarantee correctness) (Yang et al., 2025; Wang & Sun, 2025). This issue is especially amplified in long-context scenarios, where different segments carry varying information value and should be compressed with different compression rates (Jiang et al., 2023; 2024; Tang et al., 2025a). Existing dynamic compression rate allocation mechanisms are limited in several ways: they either follow predefined linear rules lacking adaptability (Jiang et al., 2023; 2024; Tang et al., 2025a), rely on model-internal understanding (Chen et al., 2025) (*i.e.*, task-agnostic) that may ignore query-relevant content, or determine compression rates based solely on relevance (Cao et al., 2024) (*i.e.*, assigning lower compression to high relevance content). None of these methods account for semantic redundancy among compression units, leading to repeated retention of similar information and compromising both compression effectiveness and information diversity. This naturally raises a research question: *How can we retain query-relevant information while identifying and eliminating semantic redundancy among compressed representations, especially under high compression rates, to jointly optimize relevance and diversity?*

To this end, we propose **COMI** (**CO**arse-to-fine Context Compression via **M**arginal **I**nformation Gain), a coarse-to-fine context compression framework that adaptively balances relevance information preservation and redundancy elimination. We introduce *Marginal Information Gain (MIG)*, which is defined as the relevance of a unit to the query minus its semantic redundancy with other units. This metric jointly captures relevance and semantic uniqueness, guiding the compression process to prioritize information that is both relevant and low redundant. COMI employs a coarse-to-fine compression strategy. In the first stage, **Coarse-grained Group Reallocation**, the input context is divided into segments of equal length, each treated as a compression group. MIG is computed for inter-group, and the compression rate for each group is dynamically adjusted accordingly: groups with high MIG (*i.e.*, high relevance and low redundancy) are assigned lower compression rates. This enables the compression budget to be adaptively reallocated based on the distribution of information value in the context. In the second stage, **Fine-Grained Token Merging**, tokens are weighted by their intra-group MIG, and fine-grained semantic fusion is performed accordingly. Tokens with high MIG contribute more to the merged representation, ensuring that key semantic units are preserved while avoiding the accumulation of "relevant but redundant" content. Through this hierarchical compression mechanism, COMI effectively retains high-relevance, low-redundancy information even under high compression rates, ensuring that the final compressed representation is semantically complementary rather than redundancy, thereby increasing information diversity.

Our contributions are three-fold: (1) We introduce Marginal Information Gain (MIG) as a metric for context compression, jointly modeling task relevance and semantic redundancy. This overcomes the limitations of existing relevance-only methods and provides a more discriminative framework for evaluating information value in long context compression. (2) We propose COMI, which employs a coarse-to-fine adaptive compression strategy. At the coarse level, reallocation based on inter-group MIG dynamically adjusts compression rates across different regions. At the fine level, intra-group MIG-guided weighted fusion eliminates group redundancy, preventing the accumulation of similar content. (3) We conduct comprehensive experiments on long-context tasks including Question-Answering (QA), summarization. Experimental results demonstrate that COMI outperforms existing methods by a large margin under high compression rates, *e.g.*, with a 32x compression constraint and using Qwen2-7B as the backbone, COMI improves the Exact Match (EM) score by approximately 25-point over suboptimal baseline on NaturalQuestions.

## 2 PRELIMINARY: GMSA

GMSA (Tang et al., 2025b) introduces an issue of cross-layer semantic misalignment in the encoder-decoder based framework and addresses it via Layer Semantic Alignment (LSA), which aligns high-level summary vectors with low-level original input semantics, thereby bridging the semantic gap across different layers. Since COMI also relies on an encoder-decoder framework, we similarly employ LSA to achieve cross-layer semantic alignment (see Figure 2). Following Tang et al. (2025b), we set LSA to a single layer.

## 3 RELATED WORK

**Task-Agnostic Context Compression Methods.** Task-agnostic context compression methods typically do not incorporate the input query and aim to preserve overall semantic information for broad downstream applicability. Main approaches include: (1) Encoder-decoder methods (Ge et al., 2024; Cheng et al., 2024; Tan et al., 2024; Liao et al., 2025; Li et al., 2025; Rau et al., 2024; Dai et al., 2025; Choi et al., 2025; Tang et al., 2025b; Zhao et al., 2025a; Liu et al., 2025b), which compress input via an encoder and decode answers from the compact representation; (2) Attention mask modification (Mu et al., 2023; Petrov et al., 2025; Ye et al., 2025), learning compact soft tokens guided by task losses; (3) Autoregressive modeling (Chevalier et al., 2023; Zhang et al., 2025), treating compression as a sequential generation process conditioned on prior compressed segments; (4) Metric-driven methods using entropy (Li et al., 2023; Jiang et al., 2023) or bidirectional semantics (Pan et al., 2024) to remove less important content; and (5) Summarization-based methods (Xu et al., 2024), training a task-agnostic summarizer for compression. *Despite preserving general semantics, these methods lack awareness of query-related tokens, inevitably discarding useful information and degrading performance, especially under high compression constraints.*

**Task-Aware Context Compression Methods.** Task-aware context compression methods do not pursue the completeness of semantic over the compressed representation, but focus on retaining information relevant to downstream task. During the compression process, these methods simultaneously receive the original context and a query to merge relevant content (Cao et al., 2024), summarize (Yoon et al., 2024; Hwang et al., 2025) or delete irrelevant content (Jiang et al., 2024; Tang et al., 2025a; Liskavets et al., 2025; Fang et al., 2025; Zhao et al., 2025c). *Although these methods can filter task-relevant content, they implicitly assume conditional independence among the retained tokens, leading to significant redundancy within the preserved tokens and making it easy to mislead LLM into generating erroneous outputs.*

**KV-cache Compression.** KV-cache Compression methods compresses KV-caches layer-wise, exploring strategies like inter-layer sharing (Brandon et al., 2024), reducing heads (Shazeer, 2019; Ainslie et al., 2023), or discarding less important KVs (Xiao et al., 2024; Li et al., 2024; Zhang et al., 2025)). *The Key limitations on these methods include requiring identical compression and response models, and necessitating model-specific modifications for different KV-cache structures.*

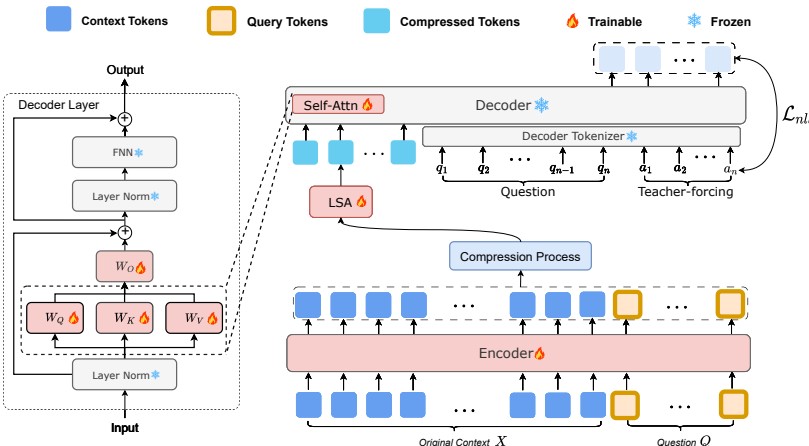

Figure 2: The Training Paradigm of COMI. COMI is based on an encoder-decoder architecture. The original context $X$ and query $Q$ are first encoded into hidden states, which are then compressed via a compression process (see Figure 3). The compressed representation is decoded and trained using cross-entropy loss. During training, the encoder and LSA are fully fine-tuned, while the decoder is partially fine-tuned, updating only the $W_Q$, $W_K$, $W_V$, and $W_O$ matrices in each layer.

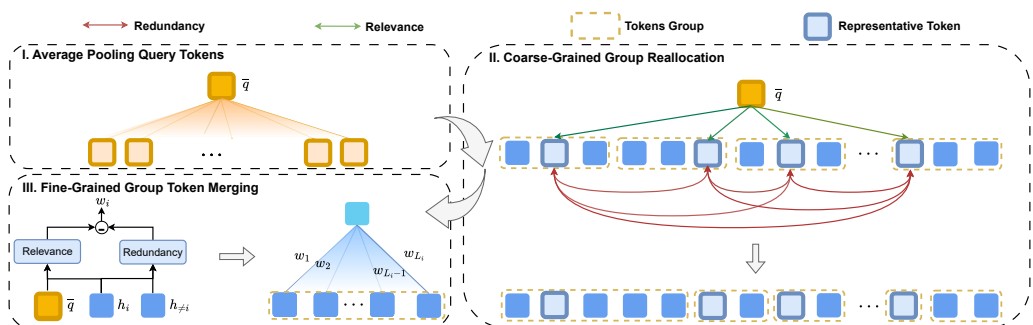

Figure 3: The Compression Process of COMI. Specifically, it sequentially performs three steps: **I. Average Pooling of Query Tokens.** Obtain a single query vector via average pooling; **II. Coarse-Grained Group Reallocation.** Reallocate the sizes of compression groups based on inter-group Marginal Information Gain (MIG) (*i.e.*, groups with higher MIG are assigned lower compression rates); **III. Fine-Grained Token Merging.** Compute the intra-group MIG for each token and merge all tokens within a group into a single compressed token according to weights $w_1, ..., w_{L_i-1}$.

# 4 COMI

In this section, we elaborate on COMI, a coarse-to-fine context compression framework based on an encoder-decoder architecture. COMI simultaneously models the relevance of each compression unit to the input question and the redundancy among units via the Marginal Information Gain (MIG). It then performs Coarse-grained Group Reallocation followed by Fine-grained Group Token Merging to retain information that is both relevant and low redundant.

## 4.1 MARGINAL INFORMATION GAIN

For a given token $x_i$, query vector $q$, and the context $X$ to which $x_i$ belongs, we compute its Marginal Information Gain (MIG) $G(x_i, q, X)$ as follows:

$$G(x_i, q, X) = \frac{x_i^\top \cdot q}{\|x_i\|\|q\|} - \max_{x_j \in X, j \neq i} \left( \frac{x_i^\top \cdot x_j}{\|x_i\|\|x_j\|} \right) \tag{1}$$

Here, the first term measures the cosine similarity between $x_i$ and the query $q$, representing its relevance; the second term captures the maximum cosine similarity between $x_i$ and any other token in $X$, reflecting its redundancy. We demonstrate that using MIG yields superior expected performance compared to relying solely on relevance. (See Appendix A)

## 4.2 COARSE-GRAINED GROUP REALLOCATION

For a given original context $X$, we first encode it via a language model (*i.e.*, encoder) to extract semantic information.

$$H = \texttt{Encoder}(X), \tag{2}$$

where $H$ is the last hidden state.

Then, we divide $H$ into $m$ *equal-length*, non-overlapping segments $H = \{S_1, S_2, \ldots, S_m\}$. Given an input query $Q = \{q_1, q_2, \ldots, q_n\}$, different segments exhibit varying degrees of relevance to $Q$ as well as differing levels of internal redundancy. Therefore, a fixed compression rate across all segments is suboptimal. Instead, we propose a dynamic compression rate reallocation strategy based on marginal information gain.

First, we average pooling the query sequence $Q$ into a single query vector $\overline{q}$:

$$\overline{q} = \frac{1}{|Q|} \sum_{q_k \in Q} q_k. \tag{3}$$

Then we select the representative vector $\hat{h}_i$ of $S_i$ that exhibits the highest relevance to the query $\overline{q}$ via:

$$\hat{h}_i = \underset{h_j \in S_i}{\operatorname{argmax}} \left( \frac{h_j^\top \cdot \overline{q}}{\|h_j\|\|\overline{q}\|} \right). \tag{4}$$

Then, we can compute the marginal information gain $G(\hat{h}_i, \overline{q}, H)$ for each segment $S_i$ as defined in Equation (1), which now evaluates the trade-off *between* the segment's relevance to the query and its redundancy with neighboring segments.

Since segments with higher marginal gain are more informative and less redundant, they should be preserved more faithfully, *i.e.*, assigned a smaller compression rate. To achieve this, we apply an inverse transformation to reverse the MIG ranking. We cant get the allocation weights via:

$$P_i = \frac{e^{-G(\hat{h}_i, \overline{q}, H)}}{\sum_{j=1}^{n} e^{-G(\hat{h}_j, \overline{q}, H)}}, \quad i = 1, 2, \ldots, n, \tag{5}$$

where $\sum_{i=1}^{n} P_i = 1$. These weights $P_i$ determine the proportion of the total allowed output length allocated to each segment.

Finally, the target length (*i.e.*, number of tokens) for the compressed representation of segment $S_i$, denoted $L_i$, is computed as:

$$L_i = L_{\text{org}} \cdot P_i, \tag{6}$$

where $L_{\text{org}}$ is the length of the original input sequence.

This dynamic reallocation mechanism enables COMI to efficiently allocate compression resources based on semantic importance and redundancy, preserving segments with higher marginal gain adaptively and improving downstream task performance under limited context length.

## 4.3 FINE-GRAINED TOKEN MERGING

After dynamically reallocating the target lengths $L_i$ for each compression segment $S_i$, we proceed to perform token merging within each segment to achieve compression. The goal is to preserve maximal informative content with respect to the query while minimizing redundancy among tokens.

For each segment $S_i = \{h_1, h_2, \ldots, h_{|S_i|}\}$, we compute the marginal information gain $G(h_k, S_i)$ for every token $x_k \in S_i$ using the same formulation as in Equation (1). This score reflects a trade-off

between the token's relevance to the query (via cosine similarity with $\overline{q}$) and its redundancy with the most similar token *within* the same segment.

To merge all tokens in $S_i$ into a single output token $\tilde{h}_i$ that preserves the most informative content, we perform MIG-weighted merging via:

$$\tilde{h}_i = \sum_{h_k \in S_i} \frac{e^{G(h_k, \overline{q}, S_i)} \cdot h_k}{\sum_{h_k \in S_i} e^{G(h_k, \overline{q}, S_i)}}. \tag{7}$$

That is, each token in the corresponding group is weighted by the softmax of its MIG, ensuring that tokens with higher information gain contribute more to the merged compressed representation $\tilde{h}_i$.

We can get the total compressed representation $\tilde{X}$ by:

$$\tilde{X} = \{\tilde{h}_1 \odot \tilde{h}_2 \odot ... \odot \tilde{h}_m\}, \tag{8}$$

where $\odot$ denotes concatenation.

### 4.4 TRAINING OBJECTIVE

We fine-tune COMI following the joint instruction-tuning approach (Lin et al., 2024), enabling it to generate correct outputs given a query and the original context. The difference lies in that we perform fine-tuning based on compressed representations, simultaneously training the encoder, LSA (to ensure the effectiveness of the compressed representations), and the decoder's $W_Q$, $W_K$, $W_V$, and $W_O$ (to ensure knowledge extraction capability of the `Decoder`) (see Figure 2).

$$\mathcal{L}_{nll} = -\sum_{i=1}^{L_a} \log p_\phi \left( a_i \mid \text{LSA}\left( \tilde{X} \right), q_1, q_2, ..., q_n, a_{<i} \right), \tag{9}$$

where $L_a$ refers to the length of ground truth; $\text{LSA}(\cdot)$ denotes the LSA module; $p_\phi(\cdot)$ is the `Decoder` probability distribution obtained after the softmax function, and $a_i$ denotes the $i$-th token in the predicted answer.

## 5 EXPERIMENTS

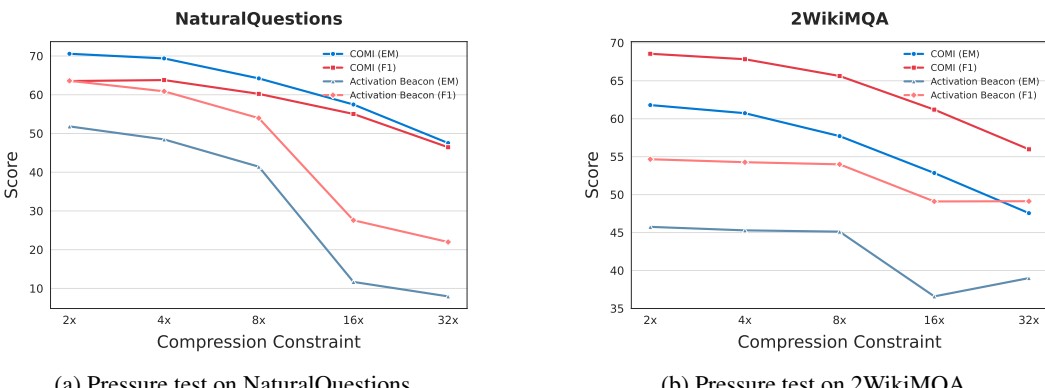

(a) Pressure test on NaturalQuestions.    (b) Pressure test on 2WikiMQA.

Figure 4: Compression Pressure Test on NaturalQuestions and 2WikiMQA. As the compression constraint increases, although both COMI and Activation Beacon generally show a downward trend in EM and F1, COMI consistently remains higher than Activation Beacon.

In this section, we seek to answer the following four research questions: (1) How does COMI perform on various tasks? (RQ1) (2) What is the impact of compression constraint on COMI's performance? (RQ2) (3) How effective is each individual component within COMI? (RQ3) (4) How does COMI impact native long context LLMs?

Table 1: Experimental results on four QA benchmarks and the MultiNews summarization dataset. We **bold** the optimal and underline the suboptimal of baselines. **EM** refers to Exact Match and **F1** refers to the F1 score. **Closed-book** indicates using only the input question as the input, while **Original Prompt** indicates using all retrieved documents as the input.

| Methods | NaturalQuestions | | 2WikiMQA | | HotpotQA | | NarrativeQA | | MultiNews |
|---|---|---|---|---|---|---|---|---|---|
| | EM | F1 | EM | F1 | EM | F1 | EM | F1 | F1 |
| *LLaMA-2-7B-Chat* | | | | | | | | | |
| Closed-book | 20.21 | 21.89 | 24.92 | 27.81 | 17.22 | 24.03 | 1.2 | 10.78 | - |
| Original Prompt | 15.04 | 26.74 | 30.82 | 37.80 | 34.34 | 47.50 | 2.16 | 13.56 | 33.2 |
| *Qwen2-7B-Instruct* | | | | | | | | | |
| Closed-book | 22.64 | 20.15 | 30.07 | 20.97 | 2.82 | 10.18 | 2.82 | 10.18 | - |
| Original Prompt | 72.35 | 38.01 | 59.78 | 47.52 | 64.17 | 53.68 | 24.53 | 9.74 | 31.42 |
| *16x compression constraint* | | | | | | | | | |
| *LLaMA-2-7B-Chat* | | | | | | | | | |
| StreamLLM | 8.06 | 18.72 | 13.42 | 15.92 | 14.45 | 21.65 | 0.00 | 0.00 | 24.66 |
| SnapKV | 11.90 | 26.49 | 25.82 | 29.55 | 26.09 | 37.87 | 0.00 | 0.00 | 19.79 |
| Activation Beacon | 4.56 | 16.58 | 2.20 | 13.84 | 11.41 | 24.68 | 6.39 | **21.16** | 26.36 |
| ICAE | 1.50 | 5.93 | 12.92 | 21.52 | 8.24 | 16.90 | 0.00 | 3.86 | 20.85 |
| LongLLMLingua | 11.64 | 23.12 | 28.42 | 32.05 | 24.75 | 35.22 | 2.44 | 15.91 | 20.66 |
| LLMLingua-2-large | 17.74 | **27.91** | 24.26 | 26.60 | 15.71 | 25.95 | 4.51 | 17.38 | 20.83 |
| GMSA | 10.36 | 15.63 | 22.68 | 28.46 | 16.09 | 25.43 | 1.13 | 9.60 | 21.10 |
| **COMI** | **22.75** | 27.32 | **41.43** | **48.74** | **36.60** | **49.50** | **8.31** | 20.32 | **27.30** |
| *Qwen2-7B-Instruct* | | | | | | | | | |
| SnapKV | 6.06 | 19.64 | 8.09 | 25.52 | 13.61 | 30.84 | 0.00 | 11.12 | 25.56 |
| Activation Beacon | 11.67 | 27.53 | 36.53 | 49.11 | 39.73 | 55.25 | 6.63 | 24.07 | 33.60 |
| LongLLMLingua | 15.07 | 29.64 | 21.80 | 29.63 | 21.58 | 35.36 | 1.03 | 11.99 | 23.16 |
| LLMLingua-2-large | 10.36 | 20.31 | 18.26 | 25.35 | 2.44 | 11.28 | 8.27 | 11.28 | 21.39 |
| GMSA | 24.41 | 28.27 | 29.49 | 34.23 | 21.54 | 30.74 | 1.88 | 10.94 | 26.55 |
| **COMI** | **56.31** | **55.52** | **52.13** | **60.43** | **45.11** | **60.52** | **13.25** | **24.69** | **35.85** |
| *32x compression constraint* | | | | | | | | | |
| *LLaMA-2-7B-Chat* | | | | | | | | | |
| StreamLLM | 4.10 | 11.86 | 9.76 | 12.07 | 8.11 | 12.97 | 0.00 | 0.00 | 19.28 |
| SnapKV | 9.91 | **24.76** | 1.53 | 2.98 | 8.25 | 13.30 | 0.00 | 0.00 | 14.44 |
| Activation Beacon | 1.47 | 11.76 | 1.71 | 13.38 | 13.94 | 26.75 | 4.79 | **17.02** | 24.35 |
| ICAE | 1.31 | 10.82 | 12.91 | 21.68 | 8.20 | 16.95 | 0.00 | 3.16 | 19.00 |
| LongLLMLingua | 7.50 | 18.86 | 28.32 | 31.76 | 25.12 | 34.78 | 2.44 | 15.10 | 18.56 |
| LLMLingua-2-large | 14.09 | 22.76 | 21.98 | 23.89 | 14.02 | 23.33 | 3.57 | 15.50 | 16.46 |
| GMSA | 9.53 | 13.94 | 24.83 | 28.62 | 17.93 | 27.15 | 1.13 | 8.56 | 19.60 |
| **COMI** | **16.99** | 23.94 | **37.34** | **43.85** | **31.46** | **43.32** | **6.83** | 15.90 | **24.71** |
| *Qwen2-7B-Instruct* | | | | | | | | | |
| SnapKV | 5.76 | 17.93 | 7.16 | 21.36 | 10.57 | 25.70 | 0.00 | 10.98 | 21.35 |
| Activation Beacon | 8.52 | 22.26 | 38.79 | 49.91 | 37.77 | 52.76 | 5.18 | 21.29 | 32.38 |
| LongLLMLingua | 13.37 | 27.39 | 21.19 | 28.91 | 22.59 | 35.45 | 1.03 | 11.91 | 21.60 |
| LLMLingua-2-large | 9.60 | 18.16 | 18.81 | 24.90 | 2.26 | 10.27 | 2.26 | 10.27 | 17.70 |
| GMSA | 24.63 | 27.81 | 28.59 | 32.81 | 21.28 | 30.08 | 2.73 | 11.04 | 25.49 |
| **COMI** | **49.15** | **49.59** | **48.89** | **57.07** | **40.46** | **55.24** | **11.18** | **22.30** | **33.60** |

## 5.1 SETTINGS

**Training.** COMI requires *only a single training run* to be effectively applied to multiple downstream tasks (*i.e.*, QA, summarization). We sampled 20,000 examples each from NaturalQuestions (Liu et al., 2024a), HotpotQA (Yang et al., 2018), 2WikiMQA (Ho et al., 2020), NarrativeQA (Kociský et al., 2018), and MultiNews (Fabbri et al., 2019) to form the final training set. All training and testing samples have token lengths no longer than 32K. During training, the batch size is set to 64, the learning rate is set to 1e-5, and a linear decay schedule is employed. The training paradigm is illustrated in Figure 2. During training, we randomly sample a compression rate (*e.g.*, 16x or

32x) for each training sample. To ensure fair comparison, we train GMSA in the same datasets and settings.

**Implementation.** Our implementation is based on LLaMA-2-7B (Chat) and Qwen2-7B (Instruct). To ensure a fair comparison, all baseline results are re-implemented using official open-source code. All experiments are conducted using the Hugging Face framework on 8 NVIDIA H20 (94GB) GPUs.

**Evaluation Metrics.** For question answering (*i.e.*, NaturalQuestions, HotpotQA, 2WikiMQA, and NarrativeQA), we report both Exact Match (EM) (Lewis et al., 2020) and F1 score (Yang et al., 2018). Summarization performance on MultiNews is measured by F1 score.

**Baselines.** We conduct comprehensive comparisons with various methods in both context compression and KV-cache compression, including hard prompt compression methods (*i.e.*, LongLLMLingua (Jiang et al., 2024), LLMLingua-2-large (Pan et al., 2024)), soft prompt compression methods (*i.e.*, ICAE (Ge et al., 2024), GMSA (Tang et al., 2025b)), and KV-cache compression methods (*i.e.*, StreamLLM (Xiao et al., 2024), SnapKV (Li et al., 2024), Activation Beacon (Zhang et al., 2025)).

## 5.2 MAIN RESULT

For RQ1, COMI demonstrates superior performance on both question answering (QA) and summarization tasks (see Table 1). On QA tasks, where input questions are typically specific and relate to a particular piece of information within a long context, COMI nearly achieves state-of-the-art results across all compression constraints and evaluation metrics. This holds true for single-hop questions (*i.e.*, NaturalQuestions), multi-hop questions (*i.e.*, HotpotQA, 2WikiMQA), and QA on extremely long texts (*i.e.*, NarrativeQA, which we test on samples with a maximum length of 32K). For instance, with a compression rate of 32x and using Qwen2-7B-Instruct as the backbone, COMI improves the Exact Match (EM) score by approximately 25 over suboptimal baseline. This highlights COMI's exceptional performance under high compression rates. The EM metric reflects the upper bound of a model's answer, while the F1 score additionally requires the model's output length to be as close to the ground truth as possible. The strong performance on both metrics indicates that COMI not only answers questions more accurately but also produces outputs that are consistent in length with the reference answers. On summarization task, the input query is often a global request (*e.g.*, "Please summarize the preceding text"). This setting requires the model to preserve and understand global information. COMI's leading performance on these tasks demonstrates its strong comprehension capabilities even without explicit relevant information, confirming its high robustness.

For RQ2, as shown in Figure 4, with the compression ratio gradually increasing (from 2x to 32x), while both COMI and Activation Beacon generally show a downward trend, COMI significantly outperforms Activation Beacon (especially in the EM metric, *e.g.*, nearly 40 point higher on the NaturalQuestions dataset under the 32x compression constraint). EM requires the generated answer to perfectly match the ground truth, indicating that compared to Activation Beacon, COMI effectively retains key information (directly related to the ground truth) across various compression rates.

## 5.3 ABLATION STUDY

To address RQ3, we conducted four ablation studies to examine how each component in COMI contributes to its performance (see Table 2): (1) Ours w/o Coarse-Grained Group Reallocation replaces dynamic grouping with a uniform partition, so every token group

Table 2: Ablation study analysis on NaturalQuestions, 2WikiMQA under 32x compression constraint using Qwen2-7B.

| Methods | NaturalQuestions | | 2WikiMQA | |
|---|---|---|---|---|
| | EM | F1 | EM | F1 |
| **Default** | **56.31** | **55.52** | **52.13** | **60.43** |
| w/o Coarse-Grained Group Reallocation | 54.54 | 54.26 | 49.82 | 58.06 |
| w/o Fine-Grained Tokens Merging | 50.81 | 50.82 | 47.88 | 56.45 |
| w/o Coarse-Grained level Redundancy | 54.59 | 52.81 | 50.14 | 58.64 |
| w/o Fine-Grained level Redundancy | 51.53 | 52.89 | 49.50 | 57.87 |

is compressed at the
same rate. (2) Ours w/o Fine-Grained Tokens Merging substitutes our information-gain-based weighting with plain average pooling. (3) Ours w/o Coarse-Grained level Redundancy reallocates groups based solely on relevance, ignoring inter-group redundancy. (4) Ours w/o Fine-Grained level Redundancy performs token merging without considering redundancy among tokens within the same group.

Removing any component causes a clear drop in all metrics, demostrating the necessity and effectiveness of each one. Eliminating Coarse-Grained Group Reallocation misallocates the compression budget and risks losing key information, whereas dropping Fine-Grained Token Merging deprives COMI of its fine-grained intra-group sensitivity and dilutes key details (both harming overall results). Disregarding redundancy at either the coarse-grained or fine-grained stage retains redundant information, weakening the compressed representation, making it harder for the model to learn, and ultimately degrading performance.

## 5.4 EFFICIENCY ANALYSIS

In this section, we analyze the computational efficiency of the COMI. By compressing the original context through coarse-grained group reallocation and fine-grained token merging, COMI significantly reduces the sequence length processed during generation, thereby lowering inference cost. The overall process consists of two main stages: compression and generation, whose floating-point operations (FLOPs) are modeled separately.

Table 3: Latency Evaluation (seconds) under $32\times$ compression constraint.

| Method | Compression | Generation | End-to-End |
|---|---|---|---|
| **NarrativeQA** | | | |
| Original Prompt | - | - | 7.04 |
| SnapKV | - | - | 2.10 |
| Activation Beacon | - | - | 3.93 |
| LongLLMLingua | 68.84 | 2.12 | 70.95 |
| GMSA | **1.57** | 0.53 | **2.10** |
| **COMI** | 2.76 | **0.50** | 3.27 |
| **MultiNews** | | | |
| Original Prompt | - | - | 8.58 |
| SnapKV | - | - | 5.03 |
| Activation Beacon | - | - | 7.94 |
| LongLLMLingua | 5.44 | 2.81 | 8.25 |
| GMSA | **0.52** | **3.34** | **3.86** |
| **COMI** | 0.66 | 3.43 | 4.09 |

The compression stage consists of two parts: first, the original context is partitioned into groups, and each group's size is dynamically adjusted based on query relevance and inter-group redundancy; second, tokens within each group are pooled via weighted to generate compressed representations.

Let $L_q$ the query length, and $L_g = \lceil L_{org}/r \rceil$ the compressed length, where $r$ is the compression rate. The FLOPs for the compression stage can be expressed as:

$$\text{FLOPs}^{comp} = F^{\text{GroupRealloc}}(L_{org}) + F^{\text{Pooling}}(L_{org}, L_c) + F^{\text{LSA}}(L_c),$$

where $L_c$ refers to compressed context length; $F^{\text{GroupRealloc}}(L)$ denotes the group reallocation cost (complexity $O(N_g \log N_g + N_g^2)$, with $L_g \ll L_{org}^2$); $F^{\text{merging}}(r, r)$ denotes the merging cost (complexity $O(r^2)$, with $r \ll L_{org}^2$). Due to their small input scales, both operations incur only lightweight overhead.

For the generation stage, assuming the answer length is $L_a$, $L_a$ forward passes are required. The FLOPs for the $i$-th forward pass depend on the input sequence length, *i.e.*, $L_c$ and query length $L_q$.

Thus, the FLOPs for each forward pass are given by:

$$\text{FLOPs}_i^{forward} = F^{\text{Decoder}}(L_c, L_q, i),$$

Combining both compression and generation stages, the total FLOPs is:

$$\text{FLOPs} = \sum_{i=1}^{L_a} \text{FLOPs}_i^{\text{forward}} + \text{FLOPs}^{comp}.$$

Whether in question-answering (QA) tasks or generation-centric summarization task, COMI achieves more than a $2\times$ end-to-end speedup over the Original Prompt at a $32\times$ compression constraint. For all compression methods, the end-to-end latency can be divided into compression and generation phases (see Table 3).[2] In contrast, the Original Prompt incurs only generation latency.

## 5.5 THE IMPACT OF COMI ON NATIVE LONG-CONTEXT LLMS

For RQ4, to evaluate the impact of COMI on models with native long-context capabilities, we train COMI using Qwen3-4B-Instruct (which natively supports a 256K context length) and use F1 as an evaluation metric. As shown in Table 4, even compared to the strong baseline of feeding the full original prompt, COMI achieves superior performance across all datasets at both $16\times$ and $32\times$ compression constraint. For instance, on the NaturalQuestions dataset at $16\times$ compression, COMI attains an F1 score of 34.79, compared to only 16.90 for the original prompt, demonstrating that COMI can enhance performance even for models with strong native long-context capabilities.

Table 4: The performance of COMI on different QA datasets (Qwen3-4B-Instruct as backbone).

| Method | NaturalQuestions | 2WikiMQA | HotpotQA | NarrativeQA |
|---|---|---|---|---|
| *16× compression constraint* | | | | |
| Original Prompt | 16.90 | 22.51 | 34.16 | 11.35 |
| **COMI** | **34.79** | **45.01** | **44.44** | **15.71** |
| *32× compression constraint* | | | | |
| Original Prompt | 16.90 | 22.51 | 34.16 | 11.35 |
| **COMI** | **28.89** | **41.19** | **39.71** | **13.25** |

## 6 CONCLUSION

We propose COMI, a coarse-to-fine context compression framework that dynamically optimizes for both task relevance and semantic diversity under high compression rates. By introducing Marginal Information Gain (MIG) (*i.e.*, a metric that penalizes redundancy while rewarding query relevance) COMI adaptively reallocates compression budgets *between* segments and performs token merging *within* groups. Extensive experiments demonstrate that COMI significantly outperforms existing methods, *e.g.*, achieving up to a 25-point EM gain under 32x compression. This work establishes MIG as a critical criterion for efficient and effective long-context modeling in LLMs.

## ETHICS STATEMENT

This work introduces COMI, an encoder-decoder based framework designed to achieve adaptive coarse-to-fine context compression through marginal information gain. The data and models used in our research are released under open-source licenses and sourced from open platforms. Although our work may have various societal impacts, it does not introduce any additional ethical concerns compared to existing text compression methods. Therefore, we believe it is unnecessary to specifically highlight any particular ethical issues here.

---

[2]For SnapKV and Activation Beacon, the latencies of the two phases can not be individually measurable due to coupling between compression and generation.

ACKNOWLEDGEMENT

This work was supported by Alibaba Group through Alibaba Research Intern Program. Libin Zheng is supported by National Natural Science Foundation of China (No. 62472455, U22B2060).

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

# A    THEORETICAL ANALYSIS: MIG VS. PURE RELEVANCE

## A.1    PRELIMINARIES

Let $X = \{x_1, \ldots, x_n\} \subset \mathbb{R}^d$ be a set of token embeddings, assumed to be zero-mean and unit-norm. Let $q \in \mathbb{R}^d$ be a query vector, also zero-mean and unit-norm, representing the target information or label. We use the cosine similarity to measure the relevance between embeddings. Specifically, for any two vectors $u, v \in \mathbb{R}^d$, $\cos(u, v) = u^\top v$.

We define the "relevance" of a token $x_i$ to the query $q$ as:

$$R(i) = \cos(x_i, q) = \frac{x_i^\top q}{\|x_i\|\|q\|}. \tag{10}$$

Since $x_i$ and $q$ are unit-norm, $R(i) = x_i^\top q$. This term captures the linear correlation between the token embedding and the query direction.

We also define the "redundancy" of a token $x_i$ with respect to a set of already selected tokens $S$ as:

$$\text{Redundancy}(i, S) = \max_{x_j \in S} \cos(x_i, x_j) = \max_{x_j \in S} \frac{x_i^\top x_j}{\|x_i\|\|x_j\|}. \tag{11}$$

Again, assuming unit norm, $\text{Redundancy}(i, S) = \max_{x_j \in S} x_i^\top x_j$. This measures how similar $x_i$ is to the most similar token already in the set $S$.

## A.2    MARGINAL INFORMATION GAIN (MIG)

Inspired by the max-relevance min-redundancy principle (Peng et al., 2005) and practical considerations in token compression, we define the Marginal Information Gain (MIG) of a token $x_i$ with respect to a set $S$ as:

$$G(i \mid S) = R(i) - \text{Redundancy}(i, S) = \cos(x_i, q) - \max_{x_j \in S} \cos(x_i, x_j). \tag{12}$$

When considering a single token $x_i$ in isolation (i.e., $S = \emptyset$), the redundancy term is conventionally set to 0, simplifying MIG to:

$$G(i) = \cos(x_i, q). \tag{13}$$

However, the true power of MIG comes when selecting tokens sequentially or when comparing them against a context. For selection purposes, we are interested in $G(i \mid S \cup \{x_i\})$, which is $G(i|S) = \cos(x_i, q) - \max_{x_j \in S} \cos(x_i, x_j)$.

## A.3    COMPARISON: PURE RELEVANCE VS. MIG

Let $X_{\text{top-}k}$ be the set of $k$ tokens with the highest relevance $R(i)$. Let $S_{\text{REL}}$ be a set of tokens selected greedily using only relevance $R(i)$. Let $S_{\text{MIG}}$ be a set of tokens selected greedily using MIG $G(i \mid S)$.

**Lemma A.1** (Information Preserved by Pure Relevance vs. MIG). *Assume we are selecting a set of $K$ tokens.*

1. *A strategy based on pure relevance $R(i)$ will select tokens that are highly correlated with the query $q$.*

2. *A strategy based on MIG $G(i \mid S)$ will select tokens that are highly correlated with $q$ but are dissimilar to tokens already selected.*

**Proof 1.** *Part 1. is direct: selecting based on $R(i)$ means prioritizing tokens with the highest $\cos(x_i, q)$.*

*Part 2. follows from the definition of $G(i \mid S)$. By subtracting $\max_{x_j \in S} \cos(x_i, x_j)$, the score $G(i \mid S)$ is naturally penalized when $x_i$ is highly correlated with existing tokens in $S$. This encourages the selection of tokens that provide "new" information relative to what has already been gathered, as opposed to reinforcing existing information.*

**Theorem 1** (Superiority of MIG under Redundancy). *Let $f(S) = I(S; y)$ be the mutual information (Dave, 2006) between the selected set of tokens $S$ and the target label $y$. Under the assumption that the token embeddings and the query are sampled from a distribution where high relevance often co-occurs with high redundancy among top-relevant tokens (a common scenario in natural language processing), and if $f(S)$ is approximately submodular (Bilmes, 2022), then a greedy selection strategy (Nemhauser et al., 1978) using MIG ($S_{MIG}$) is expected to yield a higher mutual information with the target $y$ compared to a greedy strategy using only relevance ($S_{REL}$).*

**Proof 2.** *Let's consider the scenario where there is significant redundancy among the most relevant tokens. Suppose we select $K$ tokens.*

***Pure Relevance Strategy ($S_{REL}$):*** *This strategy would greedily select tokens based on $R(i) = \cos(x_i, q)$. If several tokens $x_{i_1}, x_{i_2}, \ldots, x_{i_p}$ are all highly relevant to $q$ (i.e., $R(i_m)$ is large for $m = 1, \ldots, p$), but also highly correlated with each other (i.e., $\cos(x_{i_m}, x_{i_l})$ is large for $m \neq l$), the pure relevance strategy might select many of these redundant tokens.*

*Under the approximation that mutual information is proportional to the square of the correlation coefficient, $I(x_i; y) \propto R(i)^2$. When multiple tokens are highly correlated with each other, the additional information they contribute about $y$ diminishes. If $x_i$ and $x_j$ are selected and $\cos(x_i, x_j) = \tau$, the information from $x_j$ that is "new" with respect to $x_i$ is reduced. In a simplified Gaussian setting, this redundancy can lead to an information overlap approximately proportional to $\tau^2$ (Cover & Thomas, 2006). The pure relevance strategy does not explicitly account for this overlap, potentially leading to a "loss" in effective information captured compared to a strategy that penalizes redundancy.*

***MIG Strategy ($S_{MIG}$):*** *This strategy selects tokens based on $G(i \mid S) = \cos(x_i, q) - \max_{x_j \in S} \cos(x_i, x_j)$. When considering a token $x_i$ that is highly relevant to $q$ (large $\cos(x_i, q)$), but also highly redundant with an already selected token $x_j$ (large $\cos(x_i, x_j)$ for some $x_j \in S$), its MIG score $G(i \mid S)$ will be significantly reduced. This actively discourages the selection of such redundant tokens.*

*Therefore, when redundancy is present among top-relevant tokens, the MIG strategy is more likely to select a set of tokens that are both relevant to the query and collectively provide diverse, low redundant information. This leads to a better approximation of the true mutual information $I(S; y)$. If $f(S) = I(S; y)$ exhibits submodularity (a common assumption for information-theoretic objectives in feature selection), then a greedy approach based on marginal gain (MIG) is guaranteed to provide a $(1 - 1/e)$ approximation to the optimal set. The MIG's explicit penalization of redundancy directly addresses the information loss incurred by redundant features, thus yielding a higher expected mutual information.*

**Conclusion:** MIG provides a more robust criterion for token selection and compression compared to merely considering relevance. By explicitly penalizing redundancy with already selected tokens, MIG aims to capture a more diverse set of informative features, leading to a higher preservation of mutual information with the target query, especially in scenarios characterized by high token redundancy.

## B EXPERIMENTAL EVIDENCE: MIG VS. PURE RELEVANCE

To directly validate that Marginal Information Gain (MIG) better captures token importance than pure relevance, we conduct a diagnostic study on NaturalQuestions, decoupled from the full COMI pipeline. We treat each context segment's representative token and evaluate whether its score (MIG vs. relevance) predicts if the segment contains the ground-truth. We use AUC Score (Hanley & McNeil, 1982) as evaluation metric.

As shown in Table 5, MIG achieves a higher AUC (0.5809) than relevance (0.5423), demonstrating its superior discriminative capability for identifying answer-critical segments.

Furthermore, when retaining top segments by each metric, MIG yields significantly lower redundancy. We quantify redundancy as the average pairwise cosine similarity (excluding self-similarity) among the $K$ compressed embeddings $\{\mathbf{e}_1, \ldots, \mathbf{e}_K\}$:

$$\text{Redundancy Score} = \begin{cases} 0 & \text{if } K \leq 1, \\ \dfrac{1}{K(K-1)} \sum_{i=1}^{K} \sum_{\substack{j=1 \\ j \neq i}}^{K} \dfrac{\mathbf{e}_i^\top \mathbf{e}_j}{\|\mathbf{e}_i\| \, \|\mathbf{e}_j\|} & \text{if } K > 1. \end{cases}$$

Table 6 shows MIG consistently reduces redundancy across different retention ratios.

Table 5: AUC Score for predicting answer-containing segments on NaturalQuestions.

| Metric | AUC Score |
|---|---|
| Relevance | 0.5423 |
| **MIG** | **0.5809** |

Table 6: Redundancy of different metrics at different retention ratios.

| Metric | Retention Ratio | |
|---|---|---|
| | 25% | 50% |
| Relevance | 0.6839 | 0.6664 |
| **MIG** | **0.5673** | **0.5956** |

## C COMPREHENSIVE COMPARISON UNDER LOW COMPRESSION RATES

To comprehensively study the performance of different baselines under low compression rates, we evaluate COMI against additional baselines (including SnapKV, LongLLMLingua, and LLMLingua-2) on NaturalQuestions and 2WikiMQA using Qwen2-7B-Instruct as the backbone. As shown in Table 7, COMI consistently achieves the highest Exact Match (EM) scores across all compression rates ($2\times$ to $32\times$), demonstrating robust of COMI.

Table 7: EM scores under compression rates ($2\times$–$32\times$) on NaturalQuestions and 2WikiMQA.

| Method | Compression Rate | | | | |
|---|---|---|---|---|---|
| | $2\times$ | $4\times$ | $8\times$ | $16\times$ | $32\times$ |
| **NaturalQuestions** | | | | | |
| SnapKV | 6.01 | 5.99 | 6.25 | 6.06 | 5.76 |
| LongLLMLingua | 19.81 | 22.49 | 19.44 | 15.07 | 13.37 |
| LLMLingua-2-large | 15.52 | 14.39 | 12.77 | 10.36 | 9.60 |
| Activation Beacon | 51.83 | 48.47 | 41.43 | 11.68 | 7.95 |
| **COMI** | **70.58** | **69.38** | **64.22** | **57.48** | **47.53** |
| **2WikiMQA** | | | | | |
| SnapKV | 8.63 | 8.78 | 8.29 | 8.09 | 1.53 |
| LongLLMLingua | 29.27 | 25.61 | 22.42 | 21.80 | 21.19 |
| LLMLingua-2-large | 27.72 | 20.87 | 18.20 | 18.26 | 18.81 |
| Activation Beacon | 45.75 | 45.29 | 45.12 | 36.59 | 39.00 |
| **COMI** | **61.80** | **60.73** | **57.71** | **52.85** | **47.56** |

## D CASE STUDY OF COARSE-GRAINED GROUP REALLOCATION

COMI dynamically reallocates compression budgets based on inter-group MIG. For example, for a context of 256 tokens (A truncated sample from 2WikiMQA) with a target $32\times$ compression constraint, the initial group size is 32. MIG-guided reallocation adjusts group sizes to prioritize informative regions. As illustrated in Table 8, Segment 1 (containing key information relevant to query) receives a lower compression rate (final size = 18 vs. initial size = 32), while less informative segments expand. This adaptive behavior ensures information-dense regions are preserved with higher fidelity.

Table 8: Example of MIG-guided group reallocation (context length = 256, target 32× compression).

| Segment Index | 1 | 2 | 3 | 4 | 5 | 6 | 7 | 8 |
|---|---|---|---|---|---|---|---|---|
| MIG | 0.2227 | 0.0078 | -0.1719 | -0.4546 | -0.5583 | -0.3682 | -0.3203 | -0.2832 |
| Initial size | 32 | 32 | 32 | 32 | 32 | 32 | 32 | 32 |
| Reallo. size | 18 | 22 | 26 | 35 | 39 | 32 | 31 | 30 |

## E  SCALABILITY TO ULTRA-LONG CONTEXTS

To evaluate COMI's scalability beyond the 32K context length used in main experiments, we train and test on NarrativeQA with contexts up to 64K tokens using Qwen3-4B-Instruct as the backbone. As shown in Table 9, COMI maintains strong performance under extreme lengths: at 16× compression, it achieves 22.79, more than double the original prompt (10.69). This demonstrates COMI's scalability to ultra-long input scenarios.

## F  DATASETS

**NaturalQuestions.** NaturalQuestions (Liu et al., 2024a) is a large-scale dataset designed to evaluate open-domain question answering systems. It is based on real-world Google search queries and uses Wikipedia articles as its knowledge source. Unlike many other datasets, both the questions and answers in NQ are derived from authentic user behavior rather than being manually authored. Each question is paired with a complete answer, which can be either a short text span from a Wikipedia page (the "short answer") or a longer text passage (the "long answer"), enabling the simultaneous assessment of a model's precise extraction ability and its comprehension of long documents.

Table 9: F1 scores on NarrativeQA with 64K max length using Qwen3-4B-Instruct as backbone.

| Method | Compression Rate | |
|---|---|---|
| | 16× | 32× |
| Original Prompt | 10.69 | 10.69 |
| **COMI** | **22.79** | **20.80** |

The specific version we utilize contains 20 documents in total, with only one being the ground-truth document and the others serving as distractors.

**HotpotQA.** HotpotQA (Yang et al., 2018) is a dataset for multi-hop question answering. Its unique characteristic is that each question necessitates finding and synthesizing information from multiple Wikipedia articles. The questions often involve several entities and facts, requiring models to perform cross-document reasoning and linking to construct a coherent and complete answer, rather than simply extracting a single piece of information from one document.

**2WikiMQA.** 2WikiMQA (Ho et al., 2020) is another dataset specifically engineered for complex, multi-hop question answering. Similar to HotpotQA, it requires models to reason and integrate information from multiple Wikipedia documents. However, its questions typically involve more complex logic and inference chains, challenging models to not only identify relevant facts but also to understand their relationships (*e.g.*, such as comparing, contrasting, or inferring causality) to generate an accurate response.

**MultiNews.** MultiNews (Fabbri et al., 2019) is a dataset for multi-document summarization. It contains a large collection of news clusters, with each cluster composed of multiple articles reporting on the same event. The objective is to teach models how to extract key information from these disparate reports and synthesize it into a concise, coherent, and low redundant summary. This task requires models to identify redundant information, integrate knowledge from multiple sources, and generate fluent text.

**NarrativeQA.** NarrativeQA (Kociský et al., 2018) is a dataset designed to evaluate machine comprehension and summarization capabilities. It includes full-length novels and movie scripts from Project Gutenberg, paired with naturally-occurring, human-generated questions and answers. The queries often require a deep understanding of plot development, character relationships, and event

sequences, compelling models to perform sophisticated inference over long-form narratives to produce accurate responses.

## G  LANGUAGE MODEL USAGE STATEMENT

During the preparation of this manuscript, we utilize a large language model as a writing assistant. Its primary role is to refine and polish our paper, including the descriptions of our methodology and the presentation of mathematical derivations. This is done to improve the overall clarity, precision, and readability of the paper. All core ideas, experimental designs, and results are original work of the authors.

