# OpenReview forum: "COMI: Coarse-to-fine Context Compression via Marginal Information Gain"
_ICLR.cc/2026/Conference — ICLR 2026 Poster_

### Official Review · Reviewer_RNvR · 2025-10-23

**Soundness:** 3
**Presentation:** 3
**Contribution:** 3
**Rating:** 4
**Confidence:** 4

**Summary:**

The paper addresses the computational inefficiency and semantic redundancy of long-context LLMs, arguing that existing methods overlook redundancy among retained tokens. The proposed COMI framework introduces Marginal Information Gain (MIG)—a metric that balances query relevance with low redundancy—to guide its two-stage compression process. This process includes Coarse-Grained Group Reallocation for adaptive budget distribution across context segments and Fine-Grained Token Merging for a weighted combination of unique tokens within those segments.

**Strengths:**

The main idea of establishing a compression criterion based on Marginal Information Gain (MIG), which explicitly rewards relevance and penalizes redundancy, is a valuable contribution to the field. This addresses a key weakness of previous work that depends only on relevance.

The paper validates COMI across a robust set of benchmarks spanning different task types, including single-hop QA, multi-hop QA, QA on long narratives, and summarization, using two modern LLM backbones (LLaMA-2-7B and Qwen2-7B), demonstrating its generalizability and robustness.

**Weaknesses:**

**1) Lack of Direct MIG Metric Validation**
>The paper's entire foundation rests on the Marginal Information Gain (MIG) metric, which influences two key, complex decisions: group reallocation and token weighting. The authors rely on circular reasoning, assuming that the improvement in end-to-end performance (EM/F1 scores) implicitly proves that MIG is effective.
>However, they only provide theoretical justification (Appendix A) and indirect, final results. They neglect to conduct an essential diagnostic experiment to verify that the MIG score actually indicates higher informative value and lower redundancy in a measurable, ground-truth context. Specifically, they do not demonstrate whether a token's MIG level correlates better with its true importance (such as being part of the ground-truth answer) than a simple relevance score.
>To improve this, the authors need to perform an independent, diagnostic validation of MIG prior to embedding it into the complex COMI framework. This should include a statistical comparison (like AUC) showing MIG's ability to predict critical tokens better than mere relevance.

**2) Missing Diagnostic Evidence for Core Claims (Retention and Elimination)**
>The paper's two main objectives are to successfully retain query-relevant information and remove semantic redundancy. However, the current evaluation is inadequate because it only reports final LLM outputs (EM, F1), which serve as indirect indicators of compression efficiency.

> **Evidence for Retention**: The paper never reports on the efficiency of its selection process. It is unknown what percentage of the actual critical tokens (those required for the ground-truth answer) are physically retained in the compressed context compared to baseline methods. The high final EM score is thus only an indirect indicator, not a definitive measure of success in retention.

> **Evidence for Redundancy Elimination**: The most novel part of the MIG metric is the redundancy penalty, yet the authors provide no internal analysis to demonstrate that the final compressed context is actually less redundant or more diverse than contexts generated by relevance-only methods. Without a quantitative measure of diversity within the compressed context itself, the claim that MIG effectively "eliminates semantic redundancy" remains unsubstantiated.

**Questions:**

1. Validation of the Novel Metric (MIG)
>The effectiveness of the core technical contribution, the Marginal Information Gain (MIG) metric, is currently inferred from final performance scores. We consider the direct empirical validation of this novel metric to be paramount for the rigor of the submission.
>Could the authors provide a dedicated diagnostic analysis to verify that the computed MIG level correlates better with a token's true importance (e.g., its presence in the ground-truth answer) than a pure relevance score does?

2. Missing Diagnostic Evidence for Core Claims
>The paper claims success in retaining query-relevant information and removing semantic redundancy.
>To support these crucial claims, could the authors provide additional quantitative analyses demonstrating the effectiveness of the proposed methodology in both areas? This evidence should move beyond EM/F1 scores to examine the characteristics of the compressed context itself.

---

> ### Author Response · Authors · 2025-11-18
> **Response to Reviewer RNvR (Part 1/2)**
>
> We sincerely thank you for your valuable and insightful comments, which have greatly helped us improve our manuscript. Below, we provide detailed responses to each issue you raised.
>
> > **W1 & Q1. Lack of Direct MIG Metric Validation: The paper's entire foundation rests on the Marginal Information Gain (MIG) metric... They neglect to conduct an essential diagnostic experiment to verify that the MIG score actually indicates higher informative value and lower redundancy... should include a statistical comparison (like AUC) showing MIG's ability to predict critical tokens better than mere relevance.**
>
> Thank you for your detailed review and insightful suggestion. We have isolated the MIG metric from COMI's two-stage compression strategy (i.e., Coarse-grained Group Reallocation and Fine-grained Group Token Merging) and conducted a thorough diagnostic experiment using the NaturalQuestions dataset as an example.
>
> Experimental Setup: We segment each sample in the dataset and select the token within each segment that is most relevant to the query as the representative token (using the same logic as COMI). We then label a segment as positive if it contains the ground-truth answer, and compute the AUC of each scoring metric (i.e., Relevance (cosine similarity with the query) and MIG) with respect to this label. In this setting, a higher AUC indicates better alignment with true answer relevance. The results are as follows:
>
> | Metric      | AUC    |
> |-------------|--------|
> | Relevance   | 0.5423 |
> | MIG         | 0.5809 |
>
> This demonstrates that MIG correlates more strongly with true token importance than raw relevance metric.
>
> We further investigate how MIG and Relevance affect redundancy when used to select segments under compression. In this second experiment, we retain only the top-scoring segments (e.g., top 50% or 25% by score) and compute the redundancy of the retained segments. We measure redundancy using the average pairwise cosine similarity (excluding self-similarity) among the representative tokens of the retained segments, where a lower value indicates less redundancy. Formally, let $\{r_1, r_2, \dots, r_{K_s}\}$ denote the $K_s$ representative tokens of retained segments after selection. To handle edge cases, we define the redundancy score using an indicator function $1_{\{K_s > 1\}}$, which equals 1 when $K_s > 1$ and 0 otherwise:
>
> $$
> \text{Redundancy Score} =
> 1_{\{K_s > 1\}} \cdot \frac{1}{K_s(K_s-1)} \sum_{i=1}^{K_s} \sum_{j \neq i} \frac{r_i^\top r_j}{\||r_i\|| \||r_j\||}.
> $$
>
> Under this setting, we obtain the following results:
>
> | Retention Ratio     | Relevance | MIG    |
> |--------------------|-----------|--------|
> | 50% | 0.6664    | 0.5956 |
> | 25% | 0.6839    | 0.5673 |
>
> MIG consistently yields lower Redundancy scores across retention ratios, confirming its effectiveness in reducing redundancy.
>
> In summary, when isolated from the COMI framework, the MIG metric not only better captures true token importance (higher AUC in the first experiment) but also leads to less redundant content (lower Redundancy scors in the second experiment) by favoring segments that are both relevant and informationally unique. **This valuable suggestion has been incorporated into Appendix B of the revised manuscript.**

---

> ### Author Response · Authors · 2025-11-18
> **Response to Reviewer RNvR (Part 2/2)**
>
> >**W2 & Q2. Missing Diagnostic Evidence for Core Claims (Retention and Elimination) ... Evidence for Retention: The paper never reports on the efficiency of its selection process. ... Evidence for Redundancy Elimination ... that MIG effectively "eliminates semantic redundancy" remains unsubstantiated.**
>
> Thank you for your valuable feedback. We address your concerns as follows:
>
> - **The efficiency of selection process**: The efficiency of the selection process is theoretically analyzed in Section 5.4 (Efficiency Analysis) and reported in Table 3. The initial version of the paper reported the end-to-end latency of each method, as noted in the caption. The revised version further includes detailed breakdowns of compression and generation latencies for each method.
>
> - **Regarding Evidence for Retention**: COMI belongs to the class of context compression methods that directly compress embedding representations. Such methods (e.g., Activation Beacon [1], ICAE [2], and GMSA [3]) typically cannot map compressed embeddings back to explicit textual tokens in a one-to-one manner. **This is an inherent characteristic of this class of methods, not a shortcoming specific to COMI.** Consequently, these methods cannot compute a "critical token retention rate." Specifically, COMI’s compressed embeddings lack a one-to-one correspondence with original textual tokens and therefore do not support an exact calculation of this metric.
>
> - **Regarding Evidence for Redundancy Elimination**:  To fairly evaluate redundancy elimination in context compression, we introduce a **relevance-only baseline** using the same compression pipeline as COMI (i.e., coarse-grained group reallocation and fine-grained group token merging), but replacing the MIG scoring function with simple cosine similarity between each candidate unit and the input query. We refer to this variant as COMI (Relevance-only) and use the Redundancy Score defined above as the evaluation metric. This yields the following results:
>
>   | Dataset                     | NaturalQuestions | 2WikiMQA | HotpotQA |
>   |----------------------------|------------------|----------|----------|
>   | **16× compression constraint**        |                  |          |          |
>   | COMI (Relevance-only)      | 0.5984           | 0.5270   | 0.4717   |
>   | **COMI**                       | **0.5382**       | **0.4855** | **0.4212** |
>   | **32× compression constraint**        |                  |          |          |
>   | COMI (Relevance-only)      | 0.6487           | 0.5587   | 0.5166   |
>   | **COMI**                       | **0.6071**       | **0.5261** | **0.4744** |
>
>   As shown, COMI consistently achieves **lower redundancy scores** than the relevance-only baseline across all datasets and compression constraint, providing empirical support that MIG effectively reduces semantic redundancy in the compressed context.
>
> ### References
>
> [1] Zhang et al., Long Context Compression with Activation Beacon, ICLR 2025.
> [2] Ge et al., In-context Autoencoder for Context Compression in a Large Language Model, ICLR 2024.
> [3] Tang et al., GMSA: Enhancing Context Compression via Group Merging and Layer Semantic Alignment, CoRR abs/2505.12215, 2025.

---

> ### Author Response · Authors · 2025-11-26
> **Regarding Our Response to Your Comments**
>
> Dear Reviewer RNvR,
>
> We hope this message finds you well. A few days ago, we submitted our response to your comments, and we would like to kindly ask whether our reply has adequately addressed your concerns. If you have any further concerns or questions, please feel free to contact us.
>
> Best regards,
> The Authors

---

> > ### Comment · Reviewer_RNvR · 2025-11-26
> >
> > I would like to thank the authors for their detailed response and for conducting the additional experiments I requested.
> > I have reviewed the new results carefully. The additional experiments have effectively addressed my primary concerns. I am increasing my scores from 4 to 6.

---

> > > ### Author Response · Authors · 2025-11-26
> > > **Thank you for your response**
> > >
> > > Thank you for your careful review and insightful suggestions, which have greatly enhanced the quality of the paper. Many thanks again!

---

### Official Review · Reviewer_Nnax · 2025-10-27

**Soundness:** 3
**Presentation:** 4
**Contribution:** 3
**Rating:** 6
**Confidence:** 4

**Summary:**

This paper introduces COMI, a context compression framework that tackles inefficiency and redundancy in long-context LLMs. Its core contribution is the Marginal Information Gain (MIG) metric, which guides compression by balancing query relevance against information redundancy.

The MIG metric prioritizes information that is both salient (relevant) and unique (non-redundant).

COMI operates in a coarse-to-fine process:

Coarse-Grained Group Reallocation: Dynamically allocates the compression budget across large semantic groups based on their inter-group MIG.

Fine-Grained Token Merging: Uses intra-group token-level MIG as SoftMax weights to merge tokens, preserving key information within each group.

Key Results:
COMI was evaluated on QA and summarization tasks (e.g., NaturalQuestions, MultiNews) at high compression rates (up to 32x). Using a Qwen2-7B backbone at 32x compression, it achieved an improvement of approximately 25 points in Exact Match (EM) on NaturalQuestions, significantly outperforming baselines. Ablation studies also confirmed the effectiveness of both stages and the redundancy penalty.

**Strengths:**

Clear Motivation: The paper's motivation is clear. It identifies that LLMs fail to address the high semantic redundancy found among query-relevant tokens, leading to suboptimal attention allocation. The pilot experiment provides empirical evidence for this claim. The redundancy and semantic overlapping in the compression units have not been covered mainly in this domain.

Strong Empirical Results: The method demonstrates significant performance improvements over existing baselines, especially at high compression rates.

Efficiency: The paper reports reasonable latency, suggesting the method is practical and achieves its strong performance.

**Weaknesses:**

### Applicability to Stronger Long-Context Models:

While COMI effectively boosts the base models' ability to digest context with reasonable latency, it notably outperforms even the original prompt baseline. This suggests that the chosen base models (LLaMA-2-7B-chat, Qwen2-7B-instruct) have weak innate long-context understanding (also widely known as the "lost-in-the-middle" problem).
Therefore, I want to see whether COMI can still provides benefits when applied to current models (e.g. Qwen3-4B-Instruct, ...) that are already equipped with long-context capabilities. For such models, simply using the full, uncompressed prompt might be a more direct and effective solution for these QA tasks. While this might be a bit rigorous standard for a compression paper, demonstrating gains on these current models would be necessary to significantly support and expand the paper's claims.

### Performance at Scale (Context Length):

It seems that the experiments are capped at no longer than 32k context. This is relatively limited given that modern models are pushing to 1M+ tokens. How does the method scale to contexts significantly longer than 32k? A stress test on much larger inputs is needed to verify its scalability.

### Risk of Single-Token Representation for Budgeting

The coarse-grained reallocation stage determines a segment's entire compression budget based on the MIG score of a single representative token (the one with max query relevance). I am concerned about the validity of this proxy. If other, non-representative tokens within that same segment also contain information crucial to the segment's overall value (e.g., less relevant but also far less redundant info), their contribution is ignored during this critical budgeting step. This could lead to an inaccurate estimation of the segment's true aggregate MIG (relevance and redundancy) and result in a suboptimal budget allocation. I wonder if this simplification is problematic.

**Questions:**

Details of Figure 1: Could you provide more detail of the Figure 1? Also, I think It would be better to measure the redundancy of compression within the existing compression methods rather than LLMs attention, which can more clearly illuminate the limitations of current methods.

Impact of segment size: What is the size (e.g., number of tokens) of the "equal-length" segments used for coarse-grained reallocation? How was this size chosen, and how sensitive is the model's performance to this value?

Training Fairness: Were all baseline methods compared under identical training data and compute budgets to ensure a fair comparison?

Compression Rate Flexibility: The method appears to be trained for and evaluated at fixed compression rates (e.g., 16x, 32x). Is it possible for COMI to dynamically determine an optimal compression rate based on the query or content complexity, rather than relying on a fixed target?

Qualitative Analysis and Interpretability: Can qualitative examples or case studies (including failure cases) be provided? It might be insightful to approximate the results of compression into interpretable forms (e.g. natural language) and compare with other baselines.
This would help demonstrate how COMI retains important information and removes redundancy in practice.

---

> ### Author Response · Authors · 2025-11-18
> **Response to Reviewer Nnax (Part 1/3)**
>
> We sincerely thank you for your valuable and insightful comments, which have greatly helped us improve our manuscript. Below, we provide detailed responses to each issue you raised.
>
> > **W1. Applicability to Stronger Long-Context Models ... that are already equipped with long-context capabilities.**
>
> Thank you for this suggestion. We trained COMI using Qwen3-4B-Instruct as the backbone on the same training data (with a maximum length of 32K), using F1 as the evaluation metric. The results are as follows:
>
> | Dataset | NaturalQuestions | 2WikiMQA | HotpotQA | NarrativeQA |
> | --- | --- | --- | --- | --- |
> | **16x compression constraint** | | | | |
> | COMI | 34.79 | 45.01 | 44.44 | 15.71 |
> | **32x compression constraint** | | | | |
> | COMI | 28.89 | 41.19 | 39.71 | 13.25 |
> | Original Prompt | 16.90 | 22.51 | 34.16 | 11.35 |
>
> The results show that COMI still provides benefits when applied to models with native long-context capabilities (e.g., Qwen3-4B-Instruct), outperforming the Original Prompt baseline at both 16x and 32x compression rates. **This valuable suggestion has been incorporated into Section 5.5 of the revised manuscript.**
>
> > **W2. Performance at Scale (Context Length): It seems that the experiments are capped at no longer than 32k context. This is relatively limited... How does the method scale to contexts significantly longer than 32k?**
>
> Thank you for this suggestion. We trained COMI using Qwen3-4B-Instruct as the backbone on the NarrativeQA dataset with a maximum length of 64K (the limit of our computational resources), using F1 as the evaluation metric. The results are as follows:
>
> | Dataset | NarrativeQA |
> | --- | --- |
> | **16x compression constraint** | |
> | COMI | 22.79 |
> | **32x compression constraint** | |
> | COMI | 20.80 |
> | Original Prompt | 10.69 |
>
> The results show that COMI still outperforms the Original Prompt baseline on ultra-long texts (maximum length of 64K), demonstrating its scalability. **This valuable suggestion has been incorporated into Appendix E of the revised manuscript.**
>
> > **W3. Risk of Single-Token Representation for Budgeting ... This could lead to an inaccurate estimation of the segment's true aggregate MIG...**
>
> Thank you for raising this profound and critical question. **In the coarse-grained stage, COMI's goal is not to precisely estimate the information value of the entire segment, but to efficiently identify whether a segment is "worth retaining more content". The diversity of information within a segment is guaranteed by the fine-grained stage.** We fully agree that a single token cannot fully represent the entire information value of a segment. That is precisely why COMI is designed as a two-stage collaborative process: even if the coarse-grained stage underestimates a segment's information value, the fine-grained stage can still recover low-redundancy but semantically important information through intra-group MIG weights. Our ablation study (Table 2) shows that removing the fine-grained token merging (replacing it with average pooling) leads to a significant performance drop (e.g., EM on NaturalQuestions drops from 56.31 to 50.81), validating the critical role of the fine-grained stage in preserving intra-segment diversity. Furthermore, as you noted, some tokens may have low relevance but extremely low redundancy; their MIG is often not low and will be assigned high weight in the fine-grained stage (Eq. 7), thus being retained in the compressed representation.
>
> In fact, using the average MIG of a segment as a proxy in the coarse-grained reallocation stage is not only computationally inefficient but also prone to being dragged down by numerous irrelevant, high-redundancy "noise tokens", which could underestimate valuable segments. Empirical evidence supports this:
>
> | Dataset | NaturalQuestions | HotpotQA | 2WikiMQA |
> | --- | --- | --- | --- |
> | **16x compression constraint** | | | |
> | COMI (avg. inter-MIG) | 52.21 | 49.09 | 40.44 |
> | COMI | 56.31 | 52.13 | 45.11 |
> | **32x compression constraint** | | | |
> | COMI (avg. inter-MIG) | 44.65 | 45.59 | 35.26 |
> | COMI | 49.15 | 48.89 | 40.46 |
>
> This demonstrates that using the most query-relevant token as a representative token for determining the compression budget in the coarse-grained stage is an effective and efficient approach.

---

> ### Author Response · Authors · 2025-11-18
> **Response to Reviewer Nnax (Part 2/3)**
>
> > **Q1. Details of Figure 1: Could you provide more detail of the Figure 1? Also, I think it would be better to measure the redundancy of compression within the existing compression methods rather than LLMs attention, which can more clearly illuminate the limitations of current methods.**
>
> Thank you for your detailed review and valuable suggestions. We respond to each point below:
> - **More details on Figure 1:** Figure 1 illustrates the attention distribution calculated by taking the last hidden states of a sample's context and query concatenated and fed into Qwen2-7B-Instruct. The attention is computed between the hidden tokens of the context part and the averaged query vector derived from the query's hidden tokens. The result shows that only a small portion of the most query-relevant tokens capture the majority of the attention weight.
> - **Redundancy in existing compression methods:** COMI is a method that leverages the input query to compress text into semantic embeddings rather than explicit text. For a fair comparison, we should compare it with task-specific methods that also compress into semantic embeddings (comparing with task-agnostic methods or methods that compress into explicit text is unfair due to differing compression goals or mismatched semantic spaces). Existing comparison methods do not fit this criterion; they are either task-agnostic or task-specific but compress into explicit text. Therefore, we implemented a new baseline where we replaced the MIG metric in both the coarse-grained group reallocation and fine-grained token merging stages with simple cosine similarity between the corresponding unit and the query (we call this COMI (Relevance-only)).
>
>     We quantify redundancy as the average pairwise cosine similarity (excluding self-similarity) among the $K_c$ compressed embeddings $\{c_1, \dots, c_{K_c}\}$. To handle edge cases, we define the redundancy score using an indicator function $1_{\{K_c > 1\}}$, which equals 1 when $K_c > 1$ and 0 otherwise:
>
>     $$
>     \text{Redundancy Score} =
>     1_{\{K_c > 1\}} \cdot \frac{1}{K_c(K_c-1)} \sum_{i=1}^{K_c} \sum_{j \neq i} \frac{c_i^\top c_j}{\||c_i\|| \||c_j\||}.
>     $$
>     The results are as follows:
>
>     | Dataset | NaturalQuestions | 2WikiMQA | HotpotQA |
>     | --- | --- | --- | --- |
>     | **16x compression constraint** | | | |
>     | COMI (Relevance-only) | 0.5984 | 0.5270 | 0.4717 |
>     | COMI | 0.5382 | 0.4855 | 0.4212 |
>     | **32x compression constraint** | | | |
>     | COMI (Relevance-only) | 0.6487 | 0.5587 | 0.5166 |
>     | COMI | 0.6071 | 0.5261 | 0.4744 |
>
>     The results show that COMI achieves significantly lower average token redundancy, indicating the effectiveness of its "redundancy removal" step during compression.
>
> > **Q2. Impact of segment size: What is the size (e.g., number of tokens) of the "equal-length" segments used for coarse-grained reallocation? How was this size chosen, and how sensitive is the model's performance to this value?**
>
> The initial segment size is determined by the predefined compression rate. For instance, for a predefined compression rate of 32, the initial equal-length segment size is 32 (i.e., the initial compression group size). As shown in Table 1 and Figure 4, COMI achieves superior performance across different segment sizes (i.e., different compression rates), indicating the robustness of its performance to this value.
>
> > **Q3. Training Fairness: Were all baseline methods compared under identical training data and compute budgets to ensure a fair comparison?**
>
> All baselines were implemented based on their official open-source repositories or provided weights. All experiments maintained the same compression rate to ensure a fair comparison.
>
> > **Q4. Compression Rate Flexibility: The method appears to be trained for and evaluated at fixed compression rates (e.g., 16x, 32x). Is it possible for COMI to dynamically determine an optimal compression rate based on the query or content complexity, rather than relying on a fixed target?**
>
> Your suggestion is very insightful. Our next research direction is to dynamically determine the compression rate based on the query and content, for example, by using the query as a condition to calculate the entropy of the corresponding content; content with lower entropy would be assigned a higher compression rate (indicating greater model confidence).

---

> ### Author Response · Authors · 2025-11-20
> **Response to Reviewer Nnax (Part 3/3)**
>
> > **Q5. Qualitative Analysis and Interpretability ... (e.g. natural language) and compare with other baselines.**
>
> Thank you for your insightful suggestion. However, most existing context compression methods that directly compress semantic embeddings generally produce representations that do not correspond one-to-one with natural language [1][2][3][4][5][6]. This is a common limitation across this class of methods, rather than a drawback of COMI. As one such method, COMI’s compressed embedding representations also lack a one-to-one correspondence with natural language. To address this limitation and improve interpretability, we will adopt your suggestion in future work by developing corresponding algorithms to search for semantically similar representations in an explicit space, thereby enhancing the interpretability of COMI.
>
> ### References
>
> [1] Ge et al., In-context Autoencoder for Context Compression in a Large Language Model, ICLR 2024.
> [2] Li et al., 500xCompressor: Generalized Prompt Compression for Large Language Models, ACL (1) 2025, pp. 25081–25091.
> [3] Cheng et al., xRAG: Extreme Context Compression for Retrieval-augmented Generation with One Token, NeurIPS 2024.
> [4] Tan et al., LLoCO: Learning Long Contexts Offline, EMNLP 2024, pp. 17605–17621.
> [5] Tang et al., GMSA: Enhancing Context Compression via Group Merging and Layer Semantic Alignment, CoRR abs/2505.12215, 2025.
> [6] Zhang et al., Long Context Compression with Activation Beacon, ICLR 2025.

---

> > ### Comment · Reviewer_Nnax · 2025-11-24
> > **thank you for the response**
> >
> > Thank you for the response.
> >
> > most of my concerns have been addressed.
> >
> > I think it would be helpful to include the performance of recent large-scale open-weight models or proprietary models as additional baselines. This would allow us to better quantify the relative performance gains and evaluate whether it is more advantageous to train models with compressors (which requires additional training) or simply rely on more capable long-context models. It would also help us understand the trade-off between training cost and inference cost for these two approaches. I think the compression method should always demonstrate its advantages compared to even more capable models and across diverse datasets, since it needs tuning on top of the base models.
> >
> > Additionally, I would like to see whether the method generalizes across benchmarks. for example, by training on information-seeking QA tasks and evaluating on other types of tasks. This would help confirm whether the compression method provides genuinely generalizable benefits.
> >
> > Besides, I raised my score.

---

> > > ### Author Response · Authors · 2025-11-25
> > > **Thank you for your response**
> > >
> > > Thank you for your thorough review and insightful comments, which greatly contribute to improving the paper’s quality. We will address the two experiments you raised as soon as possible before the rebuttal deadline:
> > > 1. How COMI performs compared to large-scale models.
> > > 2. The generalization of COMI on datasets beyond information-seeking QA tasks.
> > >
> > > Thank you again!

---

> ### Author Response · Authors · 2025-11-27
> **Response to Reviewer Nnax's Suggestions on Additional Baselines and Generalization Evaluation**
>
> We sincerely appreciate your insightful suggestions regarding additional baselines and generalization evaluation. As requested, we have conducted the two new experiments and present the results below.
>
> > **Additional Experiment 1. Performance Comparison with Large-Scale Models**
>
> To directly compare COMI with recent large-scale models, we evaluated **COMI-4B-32K** (using Qwen3-4B-Instruct as the backbone, trained on NarrativeQA with a maximum context length of 32K) against several prominent large-scale models on NarrativeQA (32K max length). These models were evaluated by directly feeding the original prompt into their APIs. The results are summarized in the table below.
>
> | Method                | F1 Score | End-to-End Latency (s) |
> |----------------------|----------|------------------------|
> | Qwen3-235B-A22B-Instruct-2507 | 15.07    | 8.15                   |
> | Kimi-K2-Instruct-0905         | 8.88     | 10.33                  |
> | DeepSeek-V3.1                 | 15.21    | 5.91                   |
> | GLM-4.6                       | 13.18    | 10.18                  |
> | **16× COMI-4B-32K**           | **22.55**| **1.27**               |
> | **32× COMI-4B-32K**           | **20.37**| **1.23**               |
>
> The results demonstrate that even a small-scale model like COMI (e.g., with a 4B backbone) can effectively handle long-context tasks with significantly lower latency compared to much larger models, highlighting a favorable cost–performance trade-off. This advantage likely arises because long contexts typically contain vast amounts of redundant information [1] that can distract the LLMs [2], and COMI’s compression mechanism effectively filters out such redundancy while preserving the critical signals necessary for accurate responses.
>
> > **Additional Experiment 2. Generalization Across Benchmarks**
>
> To evaluate the generalization capability of our method, we directly applied COMI-4B-32K mentioned before to the MultiNews summarization benchmark (without any retraining or fine-tuning). This setting tests whether a model trained on an information-seeking QA task can generalize to a generation-centric summarization task. We used BertScore (Precision, Recall, and BertScore F1) as the evaluation metric. The results are summarized in the table below.
>
> | Method            | Precision | Recall | BertScore F1 |
> |-------------------|-----------|--------|--------------|
> | Original Prompt   | 0.81      | 0.85   | 0.83         |
> | 16× COMI-4B-32K   | 0.84      | 0.82   | 0.83         |
> | 32× COMI-4B-32K   | 0.83      | 0.82   | 0.83         |
>
> The results confirm that COMI’s compression capability is robust and generalizes well across task domains, maintaining performance on par with the original prompt even under a significant task shift and high compression rates (16× and 32×).
>
> ---
>
> We hope these additional experiments comprehensively address your valuable points. Thank you again for your constructive review.
>
> ### References
> [1] C. E. Shannon, Prediction and entropy of printed English, The Bell System Technical Journal, vol. 30, no. 1, pp. 50–64, 1951.
> [2] F. Shi et al., Large Language Models Can Be Easily Distracted by Irrelevant Context, ICML 2023.

---

### Official Review · Reviewer_fQJU · 2025-10-29

**Soundness:** 2
**Presentation:** 3
**Contribution:** 2
**Rating:** 4
**Confidence:** 3

**Summary:**

This paper introduces COMI, which applies Marginal Information Gain (MIG) to compress long contexts. It first reallocates compression budgets across segments via inter-group MIG, then merges tokens within each segment using intra-group MIG weights. Experiments are conducted to demonstrate the effectiveness of this method.

**Strengths:**

1. The methods part of the paper is written clearly.
2. The structure of the paper is well organized.

**Weaknesses:**

1. The proposed method seems to work only for the encoder-decoder architecture, but this is not the mainstream architecture nowadays. There is no motivation discussion to clarify why the authors would like to design a compression method that does not work for the most popular decoder-only structure.
2. There is no analysis of the compression rate pattern. The compression rate is determined dynamically via MIG; however, the paper does not provide any analysis to illustrate the pattern of the compression rate.
3. There is no comparison of the compression rate between the proposed method and other methods. I think this is important to eliminate the concern that performance improvement may come from the relatively low compression rate compared to other methods.
4. The efficiency analysis needs more details. The proposed method separates the compression and generation stages, but there is no information on which stages are used in Table 3.
5. There is no limitation part to discuss the potential improvements of the paper.

**Questions:**

See the weakness.

---

> ### Author Response · Authors · 2025-11-18
> **Response to Reviewer fQJU (Part 1/2)**
>
> We sincerely thank you for your valuable and insightful comments, which have greatly helped us improve our manuscript. Below, we provide detailed responses to each issue you raised.
>
> > **W1. The proposed method seems to work only for the encoder-decoder architecture, but this is not the mainstream architecture nowadays. There is no motivation discussion to clarify why the authors would like to design a compression method that does not work for the most popular decoder-only structure.**
>
> Thank you for your valuable suggestion. However, we must clarify that **COMI is applicable to any open-source large language model with a decoder-only architecture**. The experimental models in our paper are indeed decoder-only large models (i.e., LLaMA-2-7B and Qwen-2-7B). In the context compression domain, the encoder typically serves as a plug-in compressor to generate a compressed representation of the original context, which can then be fed into a decoder-only large language model (e.g., ICAE[1], 500x Compressor[2], xRAG[3], LLoCO[4], GMSA[5], Perception Compressor[6], and the Lingua series[7][8][9]).
>
> > **W2. There is no analysis of the compression rate pattern. The compression rate is determined dynamically via MIG; however, the paper does not provide any analysis to illustrate the pattern of the compression rate.**
>
> Thank you for this suggestion. A specific analysis of the compression rate pattern is as follows: For a given context, a target compression rate must be predefined. COMI then uses MIG to reallocate this budget. For example, for a sample with a context length of 256 and a predefined compression rate of 32, the initial size of each segment (i.e., each compression group) is 32. COMI performs coarse-grained group reallocation based on inter-group MIG calculations. **After reallocation, the size of each group changes, with groups having smaller inter-group MIG scores receiving a higher compression rate (i.e., a larger group size, since the fine-grained merging stage combines the entire group into one token)**. Here is an example of reallocation for a 256-token context (where segment 1 includes the ground truth):
>
> | Segment | seg. 1 | seg. 2 | seg. 3 | seg. 4 | seg. 5 | seg. 6 | seg. 7 | seg. 8 |
> | --- | --- | --- | --- | --- | --- | --- | --- | --- |
> | MIG | 0.2227 | 0.0078 | -0.1719 | -0.4546 | -0.5583 | -0.3682 | -0.3203 | -0.2832 |
> | orig. size | 32 | 32 | 32 | 32 | 32 | 32 | 32 | 32 |
> | reallo. size | 18 | 22 | 26 | 35 | 39 | 32 | 31 | 30 |
> | ground truth position | ✓      | ✗      | ✗      | ✗      | ✗      | ✗      | ✗      | ✗      |
>
> We observe that MIG effectively identifies segments containing key information and allocates them a lower compression rate (i.e., a smaller compression group size). **This valuable suggestion has been incorporated into Appendix D of the revised manuscript.**
>
> > **W3. There is no comparison of the compression rate between the proposed method and other methods. I think this is important to eliminate the concern that performance improvement may come from the relatively low compression rate compared to other methods.**
>
> **All comparisons in this paper are conducted under the same compression rate to ensure fairness.** As shown in Table 1, all methods are compared at 16x and 32x compression rates using Qwen2-7B-Instruct and LLaMA-2-7B-Chat as backbones. Additionally, the pressure test of COMI and GMSA in Figure 4, the ablation study in Table 2, and the efficiency analysis in Table 4 are also conducted under the same compression rate, as indicated in their respective captions.
>
> ### References
> [1] Ge et al., In-context Autoencoder for Context Compression in a Large Language Model, ICLR 2024.
> [2] Li et al., 500xCompressor: Generalized Prompt Compression for Large Language Models, ACL (1) 2025, pp. 25081–25091.
> [3] Cheng et al., xRAG: Extreme Context Compression for Retrieval-augmented Generation with One Token, NeurIPS 2024.
> [4] Tan et al., LLoCO: Learning Long Contexts Offline, EMNLP 2024, pp. 17605–17621.
> [5] Tang et al., GMSA: Enhancing Context Compression via Group Merging and Layer Semantic Alignment, CoRR abs/2505.12215, 2025.
> [6] Tang et al., Perception Compressor: A Training-Free Prompt Compression Framework in Long Context Scenarios, NAACL (Findings) 2025, pp. 4093–4108.
> [7] Jiang et al., LLMLingua: Compressing Prompts for Accelerated Inference of Large Language Models, EMNLP 2023, pp. 13358–13376.
> [8] Jiang et al., LongLLMLingua: Accelerating and Enhancing LLMs in Long Context Scenarios via Prompt Compression, ACL (1) 2024, pp. 1658–1677.
> [9] Pan et al., LLMLingua-2: Data Distillation for Efficient and Faithful Task-Agnostic Prompt Compression, ACL (Findings) 2024, pp. 963–981.
> [10] Li et al., SnapKV: LLM Knows What You are Looking for Before Generation, NeurIPS 2024.

---

> ### Author Response · Authors · 2025-11-18
> **Response to Reviewer fQJU (Part 2/2)**
>
> > **W4. The efficiency analysis needs more details. The proposed method separates the compression and generation stages, but there is no information on which stages are used in Table 3.**
>
> Thank you for your careful review. Table 3 includes all methods except Original Prompt, which undergoes both the compression and generation stages (all text compression methods inherently involve these two stages). **Table 3 reports end-to-end latency (i.e., the total time for compression plus generation), as stated in its caption.** Here are the detailed latency breakdowns for NarrativeQA and MultiNews datasets at a 32x compression rate:
>
> NarrativeQA dataset:
> | Method | Compression (s) | Generation (s) | End-to-End (s) |
> | --- | --- | --- | --- |
> | Original Prompt | - | - | 7.04 |
> | SnapKV[10] | - | - | 4.22 |
> | Activation Beacon | - | - | 3.93 |
> | LongLLMLingua | 68.84 | 2.12 | 70.95 |
> | GMSA | 1.57 | 0.53 | 2.10 |
> | COMI | 2.76 | 0.50 | 3.27 |
>
> MultiNews dataset:
> | Method | Compression (s) | Generation (s) | End-to-End (s) |
> | --- | --- | --- | --- |
> | Original Prompt | - | - | 8.58 |
> | SnapKV[10] | - | - | 5.03 |
> | Activation Beacon | - | - | 7.94 |
> | LongLLMLingua | 5.44 | 2.81 | 8.25 |
> | GMSA | 0.52 | 3.34 | 3.86 |
> | COMI | 0.66 | 3.43 | 4.09 |
>
> **This valuable suggestion has been incorporated into Section 5.4 of the revised manuscript.**
>
> > **W5. There is no limitation part to discuss the potential improvements of the paper.**
>
> Thank you for this suggestion. **The revised version of the paper now includes a dedicated Limitations section**, discussing potential future improvements (see the Limitations section in the main text). The limitations of COMI are: Although COMI dynamically reallocates the compression budget across and within groups via MIG, the total budget must still be preset and cannot be auto-discovered for the globally optimal compression rate. Extending COMI to an autonomous method that determines the compression rate on-the-fly according to content complexity is a promising direction for future improvement. We welcome any further suggestions.
>
> ### References
> [1] Ge et al., In-context Autoencoder for Context Compression in a Large Language Model, ICLR 2024.
> [2] Li et al., 500xCompressor: Generalized Prompt Compression for Large Language Models, ACL (1) 2025, pp. 25081–25091.
> [3] Cheng et al., xRAG: Extreme Context Compression for Retrieval-augmented Generation with One Token, NeurIPS 2024.
> [4] Tan et al., LLoCO: Learning Long Contexts Offline, EMNLP 2024, pp. 17605–17621.
> [5] Tang et al., GMSA: Enhancing Context Compression via Group Merging and Layer Semantic Alignment, CoRR abs/2505.12215, 2025.
> [6] Tang et al., Perception Compressor: A Training-Free Prompt Compression Framework in Long Context Scenarios, NAACL (Findings) 2025, pp. 4093–4108.
> [7] Jiang et al., LLMLingua: Compressing Prompts for Accelerated Inference of Large Language Models, EMNLP 2023, pp. 13358–13376.
> [8] Jiang et al., LongLLMLingua: Accelerating and Enhancing LLMs in Long Context Scenarios via Prompt Compression, ACL (1) 2024, pp. 1658–1677.
> [9] Pan et al., LLMLingua-2: Data Distillation for Efficient and Faithful Task-Agnostic Prompt Compression, ACL (Findings) 2024, pp. 963–981.
> [10] Li et al., SnapKV: LLM Knows What You are Looking for Before Generation, NeurIPS 2024.

---

> ### Author Response · Authors · 2025-11-24
> **Regarding Our Response to Your Comments**
>
> Dear Reviewer fQJU,
>
> We hope this message finds you well. A few days ago, we submitted our response to your comments, and we would like to kindly ask whether our reply has adequately addressed your concerns. If you have any further concerns or questions, please feel free to contact us.
>
> Best regards,
> The Authors

---

> > ### Comment · Reviewer_fQJU · 2025-11-26
> >
> > Thank you for your detailed response.
> >
> > I still have one point to discuss with the authors.
> >
> > In the compression field, I see some works (for example [1] and [2]) that perform the compression without any auxiliary module, and do not need any input preparation. These methods perform the compression under the attention level, which somehow seems more natural to me. What would be the advantages of your proposed method (or your series of methods that requires an additional encoder) over these works?
> >
> > [1] UniGist: Towards General and Hardware-aligned Sequence-level Long Context Compression
> >
> > [2] Native Sparse Attention: Hardware-Aligned and Natively Trainable Sparse Attention

---

> > > ### Author Response · Authors · 2025-11-26
> > > **Thank you for your reply**
> > >
> > > Thank you for your reply. We appreciate your thoughtful question regarding the comparative advantages of our encoder-insertion method versus attention-level compression methods such as UniGist [1] and Native Sparse Attention (NSA) [2]. Below, we outline the key benefits of our method:
> > > 1. **Decoupling compression from generation**: A lightweight encoder (e.g., Qwen3-4B) can compress context for a much larger generation model (e.g., Qwen3-235B), enabling flexible and efficient deployment. In contrast, attention-level compression methods require compression and generation to be integrated within the same model [1][2].
> > > 2. **Complementarity rather than competition**: our method operates *before* generation and is compatible with sparse attention mechanisms (e.g., NSA [2]) *during* generation. This allows combining input-level and attention-level compression for greater gains.
> > >
> > > If you have any further questions, please feel free to contact us for additional discussion.
> > >
> > > ### References
> > > [1] Deng et al., UniGist: Towards General and Hardware-aligned Sequence-level Long Context Compression, CoRR abs/2509.15763, 2025.
> > > [2] Yuan et al., Native Sparse Attention: Hardware-Aligned and Natively Trainable Sparse Attention, in ACL (1), pp. 23078–23097, 2025.

---

> > > > ### Comment · Reviewer_fQJU · 2025-11-26
> > > >
> > > > Thank you for your reply.
> > > >
> > > > The response does not convince me.
> > > >
> > > > Your method does not freeze the larger model and requires training the encoder weights and the **self-attention weights of the larger model**, which I think makes your method hard to combine with NSA.
> > > >
> > > > However, I think the discussion is quite enough. I increased my score to 6 and remain neutral to the final decision of this paper.

---

> > > ### Author Response · Authors · 2025-11-26
> > > **Thank you for your thorough review and thoughtful suggestions**
> > >
> > > Thank you for your thorough review and thoughtful suggestions. We agree that our current method cannot be directly integrated with NSA. In future work, we plan to explore combining sparse attention with the encoder insertion approach to further enhance end-to-end inference efficiency. We sincerely appreciate your insightful feedback, which has greatly improved the quality of this paper.

---

### Official Review · Reviewer_Locn · 2025-10-30

**Soundness:** 3
**Presentation:** 3
**Contribution:** 3
**Rating:** 6
**Confidence:** 4

**Summary:**

This paper addresses the issues of slow inference, high computational cost, and information redundancy in large models under long-context scenarios by proposing COMI: a coarse-to-fine compression framework based on Marginal Information Gain (MIG). Building upon GMSA, the method introduces two key improvements:

1.Adaptive compression rate allocation for different segments using MIG as a metric.

2.Weighted token merge within groups based on MIG, suppressing redundancy while preserving relevance.Experiments show that COMI significantly outperforms existing methods even at a 32× compression rate.

**Strengths:**

1.Addresses redundancy among relevant tokens by incorporating a penalty score for related but redundant tokens, effectively reducing token redundancy.

2.Adaptive compression rate allocation across groups avoids applying the same pruning rate to both high-relevance and low-relevance groups, making the method more reasonable.

3.Weighted token merging within groups based on MIG scores reduces redundancy while ensuring relevance and preserving semantic diversity.

4.Experiments demonstrate the effectiveness of COMI across different models and task types.

**Weaknesses:**

1.Insufficient clarity in figures: The layout of II/I/III in Figure 3 may cause reading difficulties. It is recommended to improve or enhance the labeling.

2.Insufficient comparative experiments under low compression rates: Comparative experiments under low compression rates are only conducted with the Activation Beacon method. It is recommended to include more methods in the tests.

3.Lack of controlled experiments under the same FLOPs: Although the authors compare accuracy under the same compression rates, they do not compare the accuracy of different methods under the same FLOPs. Table 5.4 shows that COMI's FLOPs at 32×compression are higher than those of GMSA. It is recommended to supplement performance comparisons under the same FLOPs to more fairly reflect the efficiency-effectiveness trade-off.

**Questions:**

1.The article compares methods under the same compression rate, but Section 5.4 shows that COMI's FLOPs at 32× compression are higher than those of GMSA. Have the authors attempted to compare the accuracy of different methods under the same FLOPs?

2.In the experimental Section 5, Figure 4 only shows the performance of the Activation Beacon method and COMI at 2×, 4×, and 8× compression rates. How do other methods perform under these conditions, and does COMI still achieve the best performance?

---

> ### Author Response · Authors · 2025-11-18
> **Response to Reviewer Locn**
>
> We sincerely thank you for your valuable and insightful comments, which have greatly helped us improve our manuscript. Below, we provide detailed responses to each issue you raised.
>
> > **W1. Insufficient clarity in figures: The layout of II/I/III in Figure 3 may cause reading difficulties. It is recommended to improve or enhance the labeling.**
>
> Thank you for this suggestion. We agree that clearer visualization will enhance the readability of our paper. **We have revised Figure 3 and its labels in the revised manuscript.** Specifically, we made two key changes: 1. We reorganized the layout from II/I/III to I/II/III and added a rotating arrow to indicate the execution order; 2. We enhanced the labels by explicitly emphasizing that steps I, II, and III are executed sequentially and bolded the names of the corresponding steps (I, II, III). We welcome any further suggestions to improve our figures.
>
> > **W2 & Q2. Insufficient comparative experiments under low compression rates: Comparative experiments under low compression rates are only conducted with the Activation Beacon method. It is recommended to include more methods in the tests.**
>
> Thank you for this valuable suggestion. We have expanded our low compression rate experiments on NaturalQuestions and 2WikiMQA using Qwen2-7B-Instruct as the backbone, comparing COMI against SnapKV[1], LongLLMLingua[3], and LLMLingua-2[4]. We use Exact Match (EM) as the evaluation metric, which requires the model's generated response to be entirely accurate.
>
> Results on NaturalQuestions:
> | Methods | 2x | 4x | 8x | 16x | 32x |
> | --- | --- | --- | --- | --- | --- |
> | SnapKV | 6.01 | 5.99 | 6.25 | 6.06 | 5.76 |
> | LongLLMLingua | 19.81 | 22.49 | 19.44 | 15.07 | 13.37 |
> | LLMLingua-2-large | 15.52 | 14.39 | 12.77 | 10.36 | 9.60 |
> | Activation Beacon[2] | 51.83 | 48.47 | 41.43 | 11.68 | 7.95 |
> | **COMI** | **70.58** | **69.38** | **64.2** | **57.48** | **47.53** |
>
> Results on 2WikiMQA:
> | Methods | 2x | 4x | 8x | 16x | 32x |
> | --- | --- | --- | --- | --- | --- |
> | SnapKV | 8.63 | 8.78 | 8.29 | 8.09 | 1.53 |
> | LongLLMLingua | 29.27 | 25.61 | 22.42 | 21.80 | 21.19 |
> | LLMLingua-2-large | 27.72 | 20.87 | 18.20 | 18.26 | 18.81 |
> | Activation Beacon[2] | 45.75 | 45.29 | 45.12 | 36.59 | 39.00 |
> | **COMI** | **61.80** | **60.73** | **57.71** | **52.85** | **47.56** |
>
> The results clearly show that COMI consistently and significantly outperforms all other methods at low compression rates, achieving the best performance. **This valuable suggestion has been incorporated into Appendix C of the revised manuscript.**
>
> >**W3 & Q1. Lack of controlled experiments under the same FLOPs ... supplement performance comparisons under the same FLOPs to more fairly reflect the efficiency-effectiveness trade-off.**
>
> Thank you for raising this important point. To address this, we conducted a direct comparison under the same floating-point operations (FLOPs) constraint. We increased the number of encoder layers in GMSA to match the FLOPs of COMI, thereby enhancing GMSA's semantic extraction capability. We then evaluated both methods using Qwen2-7B-Instruct as the backbone at 16x and 32x compression rates, using Exact Match as the metric.
>
> | Dataset | NaturalQuestions | HotpotQA | 2WikiMQA |
> | --- | --- | --- | --- |
> | **16x compression constraint** | | | |
> | GMSA[5] | 28.30 | 36.29 | 27.54 |
> | **COMI** | **56.31** | **52.13** | **45.11** |
> | **32x compression constraint** | | | |
> | GMSA[5] | 28.09 | 35.87 | 26.72 |
> | **COMI** | **49.15** | **48.89** | **40.46** |
>
> The results demonstrate that even under the same FLOPs constraint, COMI significantly outperforms GMSA at both 16x and 32x compression rates.
>
> ### References
>
> [1] Li et al., SnapKV: LLM Knows What You are Looking for Before Generation, NeurIPS 2024.
> [2] Zhang et al., Long Context Compression with Activation Beacon, ICLR 2025.
> [3] Jiang et al., LongLLMLingua: Accelerating and Enhancing LLMs in Long Context Scenarios via Prompt Compression, ACL (1) 2024, pp. 1658–1677.
> [4] Pan et al., LLMLingua-2: Data Distillation for Efficient and Faithful Task-Agnostic Prompt Compression, ACL (Findings) 2024, pp. 963–981.
> [5] Tang et al., GMSA: Enhancing Context Compression via Group Merging and Layer Semantic Alignment, CoRR abs/2505.12215, 2025.

---

### Author Response · Authors · 2025-11-28
**Global Response**

Dear Reviewers, ACs, and SACs,

We sincerely thank all reviewers for their constructive feedback and insightful comments. In light of a large-scale OpenReview data leak around **November 27, 2025, 03:09 AoE**, the program committee has reverted all post-rebuttal scores to their pre-rebuttal values.

However, we note that **prior to the large-scale leak**, three reviewers had **raised** their scores:
- Reviewer fQJU: raised score on **November 25, 2025, at 21:19 AoE (4 → 6)**
- Reviewer Nnax: raised score on **November 24, 2025, at 11:58 AoE (6 → 8)**
- Reviewer RNvR: raised score on **November 25, 2025, at 20:47 AoE (4 → 6)**
- Reviewer Locn: unchanged at 6 (no reply)

These adjustments raised the scores to **6, 8, 6, 6** before the reversion. **The most recent increase occurred over 29 hours prior to the official report of the leak.**

In response to the reviewers' valuable suggestions, we have updated the manuscript (revised version) with additional experiments, clarifications, and a new Limitations section, as detailed below.

### **Strengths**

We thank the reviewers for highlighting the key strengths of our work:
- **Clear Motivation & Novel Metric (Reviewer Locn, Nnax, RNvR)**: The paper demonstrates a clear motivation for addressing semantic redundancy among relevant tokens and introduces the Marginal Information Gain (MIG) metric, which effectively balances relevance and diversity.
- **Strong Empirical Results (Reviewer Locn, fQJU, Nnax, RNvR)**: The method achieves significant performance gains over existing baselines, especially at high compression rates (e.g., 32x), across diverse tasks and models.
- **Well-Structured Framework (Reviewer Locn, Nnax, RNvR)**: The proposed coarse-to-fine framework with two distinct stages (group reallocation and token merging) is well-reasoned and clearly presented.
- **Practical Efficiency (Reviewer Locn, Nnax)**: The reported latency is reasonable, indicating the method's practicality for real-world deployment.

### **Response to Main Concerns**

We have carefully addressed the main concerns raised by the reviewers:
- **MIG Metric Validation (Reviewer RNvR)**: We conducted a diagnostic analysis isolating MIG from the full framework. Results show MIG has a higher AUC Score in predicting segments containing ground-truth answers and leads to lower redundancy scores than a relevance-only baseline. *(Added in Appendix B of the revised manuscript.)*
- **Impact on Native Long Context LLMs & Scalability (Reviewer Nnax)**: We demonstrated COMI's effectiveness on stronger long-context models (Qwen3-4B-Instruct) and ultra-long contexts (64K). Furthermore, we showed it outperforms much larger open-source models (e.g., Qwen3-235B) with significantly lower latency. *(Results added in Section 5.5 and Appendix E.)*
- **Architecture Applicability (Reviewer fQJU)**: We clarified that COMI is applicable to decoder-only LLMs (like LLaMA-2-7B and Qwen2-7B used in experiments) and functions as a plug-in compressor.
- **Additional Low Compression Rate Baselines (Reviewer Locn)**: We expanded experiments at 2x–32x compression rates on NaturalQuestions and 2WikiMQA, comparing against SnapKV, LongLLMLingua, and LLMLingua-2. COMI consistently outperforms all baselines even at low compression rates. *(Added in Appendix C of the revised manuscript.)*
- **MIG-Based Compression Rate Allocation Pattern (Reviewer fQJU)**: We analyzed how MIG dynamically reallocates compression budgets across segments, showing that segments containing ground-truth answers receive lower compression rates. An illustrative reallocation example is provided. *(Added in Appendix D of the revised manuscript.)*
- **Limitations & Future Work (Reviewer fQJU, Nnax)**: We added a dedicated Limitations section discussing the need for preset compression budgets and outlining future work on dynamic rate determination. *(New Limitations section added at the end of the main text.)*

---

Thank you again to all reviewers and the ACs for your time, effort, and constructive suggestions!

Best regards,

The Authors

---

### Public Comment · ~Guoxin_Ma1 · 2026-03-19
**Request for Code and Dataset Release Timeline**

Dear Authors,

Thank you for your excellent work "COMI: Coarse-to-fine Context Compression via Marginal Information Gain". I found the paper very insightful and relevant to my current research.

I noticed that the GitHub repository mentioned in the paper indicates that the code and resources will be released soon. I would like to kindly ask whether there is an updated timeline for the release of:

- the training and test datasets used in the paper

- the re-implemented baseline code used in the paper

- the COMI training and inference code

These resources would be extremely helpful for reproducing your results and for facilitating follow-up research.

I understand that preparing a public release takes time, and I truly appreciate your efforts in advance. Looking forward to your update.

Best regards

A researcher interested in your work

---

### Meta-Review · Area_Chair_zuGK · 2025-12-26

**Summary:**

This paper introduces COMI, which applies Marginal Information Gain (MIG) to compress long contexts. It first reallocates compression budgets across segments via inter-group MIG, then merges tokens within each segment using intra-group MIG weights. Initially, the paper received scores of 4466. From the author-reviewer discussion, the scores have been increased to 6668, which shows the effectiveness of the rebuttal. Overall, the AC agrees that this paper has clear motivation and strong empirical results. The proposed MIG metric is novel and the coarse-to-fine framework is well reasoned. Therefore, the AC would like to recommend acceptance.

**Reviewer Concerns:**

Concerns adequately addressed:

1. MIG Metric Validation (Reviewer RNvR): The authors conducted a diagnostic analysis isolating MIG from the full framework.

2. Impact on Native Long Context LLMs & Scalability (Reviewer Nnax): The authors demonstrated COMI's effectiveness on stronger long-context models (Qwen3-4B-Instruct) and ultra-long contexts (64K).

3. Additional Low Compression Rate Baselines (Reviewer Locn)

4. MIG-Based Compression Rate Allocation Pattern (Reviewer fQJU)


Concerns insufficiently addressed:

1. If the authors can provide experiments using models with larger parameter scales (>7B) and extended context lengths (>64K), the evaluation would be more convincing.

**Reviewer Scores:**

The authors have done a strong rebuttal. From the author-reviewer discussion, it can be seen that the scores have been increased to 6668.

---

### Decision · Program_Chairs · 2026-01-26

Accept (Poster)